# Satellite soil moisture data assimilation impacts on modeling weather variables and ozone in the southeastern US - part I: an overview

Min Huang[1], James H. Crawford[2], Joshua P. DiGangi[2], Gregory R. Carmichael[3], Kevin W. Bowman[4], Sujay V. Kumar[5], and Xiwu Zhan[6]

[1]George Mason University, Fairfax, VA, USA
[2]NASA Langley Research Center, Hampton, VA, USA
[3]The University of Iowa, Iowa City, IA, USA
[4]Jet Propulsion Laboratory, California Institute of Technology, Pasadena, CA, USA
[5]NASA Goddard Space Flight Center, Greenbelt, MD, USA
[6]NOAA National Environmental Satellite, Data, and Information Service, College Park, MD, USA

*Correspondence to*: Min Huang (mhuang10@gmu.edu)

**Abstract.** This study evaluates the impact of satellite soil moisture (SM) data assimilation (DA) on regional weather and ozone ($O_3$) modeling over the southeastern US during the summer. Satellite SM data are assimilated into the Noah land surface model using an ensemble Kalman filter approach within National Aeronautics and Space Administration's Land Information System framework, which is semicoupled with the Weather Research and Forecasting model with online Chemistry (WRF-Chem, standard version 3.9.1.1). The DA impacts on the model performance of SM, weather states and energy fluxes show strong spatiotemporal variability. Dense vegetation and water use from human activities unaccounted for in the modeling system are among the factors impacting the effectiveness of the DA. The daytime surface $O_3$ responses to the DA can largely be explained by the temperature-driven changes in biogenic emissions of volatile organic compounds and soil nitric oxide, chemical reaction rates, as well as dry deposition velocities. On a near-biweekly timescale, the DA modified the mean daytime and daily maximum 8 h-average surface $O_3$ by up to 2-3 ppbv, with the maximum impacts occurring in areas where daytime surface air temperature most strongly (i.e., by ~2 K) responded to the DA. The DA impacted WRF-Chem upper tropospheric $O_3$ (e.g., for its daytime-mean, by up to 1-1.5 ppbv) partially via altering the transport of $O_3$ and its precursors from other places as well as in-situ chemical production of $O_3$ from lightning and other emissions. Case studies during airborne field campaigns suggest that the DA improved the model treatment of convective transport and/or lightning production. In the cases that the DA improved the modeled SM, weather fields and some $O_3$-related processes, its influences on the model's $O_3$ performance at various altitudes are not always as desirable. This is in part due to the uncertainty in the model's key chemical inputs, such as anthropogenic emissions, and the model representation of stratosphere-troposphere exchanges. This can also be attributable to shortcomings in model parameterizations (e.g., chemical mechanism, natural emission, photolysis and deposition schemes), including those related to representing water availability impacts. This study also shows that, WRF-Chem upper tropospheric $O_3$ response to the DA has comparable magnitudes with its response to the estimated US anthropogenic emission changes within two years. As reductions in anthropogenic emissions in North America would benefit the mitigation of $O_3$ pollution in its downwind regions, this analysis highlights the important role of SM in

quantifying air pollutants' source-receptor relationships between the US and its downwind areas. It also emphasizes that using up-to-date anthropogenic emissions is necessary for accurately assessing the DA impacts on the model performance of $O_3$ and other pollutants over a broad region. This work will be followed by a Noah-Multiparameterization (with dynamic vegetation) based study over the southeastern US, in which selected processes including photosynthesis and $O_3$ dry deposition will be the foci.

## 1 Introduction

Tropospheric ozone ($O_3$) is a central component of tropospheric oxidation chemistry with atmospheric lifetimes ranging from hours within polluted boundary layer to weeks in the free troposphere (Stevenson et al., 2006; Cooper et al., 2014; Monks et al., 2015). Ground-level $O_3$ is a US Environmental Protection Agency (EPA) criteria air pollutant which harms human health and imposes threat to vegetation and sensitive ecosystems, and such impacts can be strongly linked or/and combined with other stresses, such as heat, aridity, soil nutrients, diseases, and non-$O_3$ air pollutants (e.g., Harlan and Ruddell, 2011; Avnery et al., 2011; World Health Organization, 2013; Fishman et al., 2014; Lapina et al., 2014; Cohen et al., 2017; Fleming et al., 2018; Mills et al., 2018a, b). Across the world, various metrics have been used to assess surface $O_3$ impacts (Lefohn et al., 2018). In October 2015, the US primary (to protect human health) and secondary (to protect public welfare including vegetation and sensitive ecosystems) National Ambient Air Quality Standards for ground-level $O_3$, in the format of the daily maximum 8 h-average (MDA8), were revised to 70 parts per billion by volume (ppbv, US Federal Register, 2015). Understanding the connections between weather patterns and surface $O_3$ as well as their combined impacts on human and ecosystem health under the changing climate is important to developing strong-enough anthropogenic emission control to meet targeted $O_3$ air quality standards (Jacob and Winner, 2009; Doherty et al., 2013; Coates et al., 2016; Lin et al., 2017).

Ozone aloft is more conducive to rapid long-range transport to influence surface air quality in downwind regions (e.g., Zhang et al., 2008; Fiore et al., 2009; Hemispheric Transport of Air Pollution, HTAP, 2010, and the references therein; Huang et al., 2010, 2013, 2017a; Doherty, 2015). In the upper troposphere/lower stratosphere regions, $O_3$ as well as water vapor is particularly important to climate (Solomon et al., 2010; Shindell et al., 2012; Stevenson et al., 2013; Bowman et al. 2013; Intergovernmental Panel on Climate Change, 2013; Rap et al., 2015; Harris et al., 2015). Ozone variability in the free troposphere can be strongly affected by stratospheric air, transport of $O_3$ that is produced at other places of the troposphere, as well as in-situ chemical production from $O_3$ precursors including nitrogen oxides ($NO_x$, namely nitric oxide, NO, and nitrogen oxide, $NO_2$), carbon monoxide (CO), methane, and non-methane volatile organic compounds (VOCs). Mid-latitude cyclones are major mechanisms of venting boundary layer constituents, including $O_3$ and its precursors, to the mid- and upper troposphere. They are active throughout the year and relatively weaker during the summer. Convection, often associated with thunderstorms and lightning, is a dominant mechanism of exporting pollution in the summertime (e.g., Dickerson et al., 1987; Hess, 2005; Brown-Steiner and Hess, 2011; Barth et al., 2012). During North American summers,

upper tropospheric anticyclones trap convective outflows and promote in-situ $O_3$ production from lightning and other emissions (e.g., Li et al., 2005; Cooper et al., 2006, 2007, 2009). It has also been shown that stratospheric $O_3$ intrusions are often associated with cold frontal passages and convection (e.g., Pan et al., 2014; Ott et al., 2016).

On a wide range of spatial and temporal scales, atmospheric weather and composition interact with land surface conditions (e.g., soil and vegetation states, topography, and land use/cover, LULC), which can be altered by various human activities
and/or natural disturbances such as urbanization, deforestation, irrigation, and natural disasters (e.g., Betts, 1996; Kelly and Mapes, 2010; Taylor et al., 2012; Collow et al., 2014; Guillod et al., 2015; Tuttle and Salvucci, 2016; Cioni and Hohenegger, 2017; Fast et al., 2019; Schneider et al., 2019). As a key land variable, soil moisture (SM) influences the atmosphere via evapotranspiration, including evaporation from bare soil and plant transpiration. The SM-atmosphere coupling strengths are overall strong over transitional climate zones (i.e., the regions between humid and arid climates) where evapotranspiration is
moderately high and constrained by SM (e.g., Koster et al., 2004, 2006; Seneviratne et al., 2010; Dirmeyer, 2011; Miralles et al., 2012; Gevaert et al., 2018). The southeastern US includes large areas of transitional climate zones, whose geographical boundaries vary temporally (e.g., Guo and Dirmeyer, 2013; Dirmeyer et al., 2013). Soil moisture and other land variables are currently measurable from space. It has been shown in a number of scientific and operational applications that satellite SM data assimilation (DA) impacts model skill of atmospheric weather states and energy fluxes (e.g., Mahfouf, 2010; de Rosnay
et al., 2013; Santanello et al., 2016; Yin and Zhan, 2018). An effort began recently to evaluate the impacts of satellite SM DA on short-term regional-scale air quality modeling. Based on case studies in East Asia, such effects are shown to vary in space and time, partially dependent on surface properties (e.g., vegetation density and terrain) and synoptic weather patterns. Also, the SM DA impacts on model performance can be complicated by other sources of model error, such as the uncertainty of the models' chemical inputs including emissions and chemical initial/lateral boundary conditions (Huang et al., 2018).

This study extends the work by Huang et al. (2018) to the southeastern US during intensive field campaign periods in the summer convective season. Modified from the approach used in Huang et al. (2018), we assimilate satellite SM into the Noah land surface model (LSM) within National Aeronautics and Space Administration (NASA)'s Land Information System (LIS), which is semicoupled with the Weather Research and Forecasting model with online Chemistry (WRF-Chem). The term "semicoupled" indicates that the SM DA within LIS influences WRF-Chem's land initial conditions. Atmospheric
states and energy fluxes from the no-DA and DA cases are compared with surface, aircraft, and satellite observations during selected field campaign periods. The WRF-Chem results are also compared with the chemical fields of the Copernicus Atmosphere Monitoring Service (CAMS), which serves as the chemical initial/lateral boundary condition model of WRF-Chem. Other sources of errors in WRF-Chem simulated $O_3$ are identified by a WRF-Chem emission sensitivity simulation and the stratospheric $O_3$ tracer output from the Geophysical Fluid Dynamics Laboratory (GFDL)'s Atmospheric Model,
version 4 (AM4). The modeling and SM DA approaches as well as evaluation datasets are first introduced in Section 2. Section 3 starts with an overview of the synoptic and drought conditions during the study periods (Section 3.1), followed by

discussions on the model responses to satellite SM DA. The SM DA impacts on $O_3$ export from the US and the potential impacts on surface $O_3$ in regions downwind of the US are included in the discussions. Results during a summer 2016 field campaign and a summer 2013 campaign are covered in Sections 3.2-3.3 and Section 3.4, respectively. Section 4 summarizes key results from its previous sections, discusses their implications and provides suggestions on future work.

## 2 Methods

### 2.1 Modeling and SM DA approaches

This study focuses on a summer southeastern US deployment (16-28 August 2016) of the Atmospheric Carbon Transport (ACT)-America campaign (https://act-america.larc.nasa.gov). One goal of this campaign is to study atmospheric transport of trace gases. Three WRF-Chem full-chemistry simulations (i.e., base, "assim", and "NEI14" in Table 1) were conducted throughout this campaign on a 63 vertical layer, 12 km×12 km (209×139 grids) horizontal resolution Lambert conformal grid centered at 33.5°N/87.5°W (Figure 1a-c). To help confirm surface SM impacts on atmospheric conditions, a complementary simulation "minus001" was also conducted in the same model grid only for selected events during this campaign (Table 1). Trace gases and aerosols were simulated simultaneously and interactively with the meteorological fields using the standard version 3.9.1.1 of WRF-Chem (Grell et al., 2005).

Version 3.6 of the widely-used, four-soil-layer Noah LSM (Chen and Dudhia, 2001) within LIS (Kumar et al., 2006) version 7.1rp8 served as the land component of the modeling/DA system used. An offline Noah simulation was performed within LIS prior to all WRF-Chem simulations for equilibrated land conditions (details in Section S1). Consistent model grids and geographical inputs of the Noah LSM were used in the offline LIS and all WRF-Chem simulations. Specifically, topography, time-varying green vegetation fraction, LULC type, and soil texture type inputs were based on the Shuttle Radar Topography Mission Global Coverage-30 version 2.0, Copernicus Global Land Service, the International Geosphere-Biosphere Programme-modified Moderate Resolution Imaging Spectroradiometer (Figure 1a-c), and the State Soil Geographic (Figure S1, upper, Miller and White, 1998) datasets, respectively.

Successful, valid retrievals of morning-time SM (version 2 of the 9 km enhanced product, generated using baseline retrieval algorithm) from NASA's Soil Moisture Active Passive (SMAP, Entekhabi et al., 2010) L-band polarimetric radiometer were assimilated into Noah within LIS. SMAP provides global coverage of surface (i.e., the top 5 cm of the soil column) SM within 2-3 days along its morning orbit (~6 am local time crossing) with the ground track repeating in 8 days. Compared to its predecessors that take measurements at higher frequencies, SMAP has a higher penetration depth for SM retrievals and lower attenuation in the presence of vegetation. Evaluation of SMAP data over North America with in-situ and LSM output suggests better data quality over flat and less forested regions (Pan et al., 2016), and previous studies have demonstrated that the SMAP DA improvements on weather variables are more distinguishable over regions with sparse vegetation (e.g., Huang

et al., 2018; Yin and Zhan, 2018). Before the DA, SMAP data were re-projected to the model grid and bias correction was applied via matching the means and standard deviations of the Noah LSM and SMAP data for each grid (de Rosnay et al., 2013; Huang et al., 2018; Yin and Zhan, 2018) during August of 2015-2019. Such bias correction reduced the dynamic
ranges of SM from the original SMAP retrievals. The Global Modeling and Assimilation Office (GMAO) ensemble Kalman filter approach embedded in LIS was applied, with the ensemble size of 20. Perturbation attributes of state variables (Noah SM) and meteorological forcing variables (radiation and precipitation) were based on default settings of LIS derived from Kumar at al. (2009).

All WRF-Chem cases, except case "minus001", were started on 13 August 2016. Atmospheric meteorological initial/lateral
boundary conditions were downscaled from the 3-hourly, 32 km North American Regional Reanalysis (NARR). Consistent with NARR, the WRF-Chem model top was set at 100 hPa, slightly above the climatological tropopause heights for the study region/month. The $0.083°×0.083°$ National Centers for Environmental Prediction (NCEP) daily sea surface temperature (SST) reanalysis product was used as an additional WRF forcing. Chemical initial/lateral boundary conditions for major chemical species were downscaled from the 6-hourly, $0.4°×0.4°×60$-level CAMS. Surface $O_3$ from CAMS is
positively biased over the eastern US referring to various observations, but major chemical species in the free troposphere are overall successfully reproduced (e.g., Huijnen et al., 2020; Wang et al., 2020). As WRF-Chem has only tropospheric chemistry, the lack of dynamic chemical upper boundary conditions is expected to introduce biases in the modeled $O_3$ throughout the troposphere, and such biases depend on the distribution of model vertical layers as well as the length of the simulation. To determine how this limitation of WRF-Chem affects its $O_3$ performance, we used the outputs (3-hourly,
$1°×1.25°×49$-level) from GFDL's AM4 (Horowitz et al., 2020) and its stratospheric $O_3$ tracer, which have been applied to other $O_3$ studies (e.g., Zhang et al., 2020). Since the second day of the simulation period, chemical initial conditions were cycled from the chemical fields of the previous-day simulation. Atmospheric meteorological and land fields were reinitialized every day at 00 UTC with NARR and the previous-day no-DA or DA LIS outputs, respectively. Each day's simulation was recorded hourly at 00:00 (minute:second) through the following 30 hours, forced by temporally constant SST
as the diurnal variation of the sea surface is typically smaller than land on large scales. Each day's WRF-Chem meteorological outputs served as the forcings of the no-DA and DA LIS simulations, which produced land initial conditions for next day's WRF-Chem simulations. The model output >6 hours since each day's initialization was analyzed for the period of 16-28 August 2016.

In all WRF-Chem simulations, key physics options applied include: the local Mellor–Yamada–Nakanishi–Niino planetary
boundary layer (PBL) scheme along with its matching surface layer scheme (Nakanishi and Niino, 2009), the Rapid Radiative Transfer Model short-/long-wave radiation schemes (Iacono et al., 2008), the Morrison double-moment microphysics, which predicts the mass and number concentrations of hydrometeor species (Morrison et al., 2009), and the Grell-Freitas scale-aware cumulus scheme (Grell and Freitas, 2014), which has also been implemented in the GMAO GEOS-

Forward Processing system (https://gmao.gsfc.nasa.gov/news/geos_system_news/2020/GEOS_FP_upgrade_5_25_1.php).

Chemistry related configurations are: the Carbon-Bond Mechanism version Z (Zaveri and Peters, 1999) gas phase chemical mechanism and the eight-bin sectional Model for Simulating Aerosol Interactions and Chemistry (Zaveri et al., 2008), including aqueous chemistry for resolved clouds. Both aerosol direct and indirect effects were enabled in all simulations.

Daily biomass burning emissions came from the Quick Fire Emissions Dataset (Darmenov and da Silva, 2015) version 2.5r1, and plume rise with a recent bug fix (suggested by Ravan Ahmadov, NOAA/ESRL, in August 2019) was applied. Emissions

of biogenic VOCs and soil NO were computed online (i.e., driven by the WRF meteorology) using the Model of Emissions of Gases and Aerosols from Nature (MEGAN, Guenther et al., 2006). It has been shown that MEGAN may overpredict biogenic VOC emissions over the study regions and tends to underpredict soil NO emissions especially in high-temperature (i.e., >30 °C) agricultural regions (e.g., Oikawa et al., 2015; Huang et al., 2017b, and the references therein). Important sources of uncertainty include: 1) the uncertainty in MEGAN's land and meteorological inputs including surface temperature

and radiation fields from WRF; and 2) that drought influences on these emissions are not well understood and represented in MEGAN, and such influences include biogenic VOC emissions being enhanced, reduced or terminated during various stages of droughts. Specifically, at the early stage of droughts when plants still have sufficient reserved carbon resources, dry conditions may promote these emissions via enhancing leaf temperature. Persistent droughts will terminate biogenic VOC emissions after the reserved carbon resources are consumed (e.g., Pegoraro et al., 2004; Bonn et al., 2019). Cloud-top-

height-based lightning parameterization was applied (Wong et al., 2013). The intra-cloud to cloud-to-ground flash ratio was based on climatology (Boccippio et al., 2001), and lightning NO was distributed using vertical profiles in Ott et al. (2010). For both intra-cloud and cloud-to-ground flashes, 125 moles of NO were emitted per flash, close to the estimates in several studies for the US (e.g., Pollack et al., 2016; Bucsela et al., 2019). The passive lightning $NO_x$ tracer was implemented, which experienced atmospheric transport but not chemical reactions. Anthropogenic emissions in the base, "assim" and

"minus001" simulations (Table 1) were based on US EPA's National Emission inventory (NEI) 2016 beta, and NEI 2014 was used in the "NEI14" simulation. The differences between NEI 2016 beta and earlier versions of NEIs, such as NEI 2014 and 2011, are summarized at: http://views.cira.colostate.edu/wiki/wiki/10197/inventory-collaborative-2016beta-emissions-modeling-platform, for various chemical species. Anthropogenic emissions of $O_3$ precursors are lower in NEI 2016 beta than in NEI 2014 (by <20% for key species) as well as NEI 2011, in which $NO_x$ emissions may be positively biased for 2013

(Travis et al., 2016). These differences are qualitatively consistent with the observed trends of surface air pollutants (https://www.epa.gov/air-trends).

Chemical loss via dry deposition (i.e., dry deposition velocity $v_d$ multiplied by surface concentration) was calculated based on the widely-used Wesely scheme (Wesely, 1989, details in Section S2). This scheme defines $v_d$ as the reciprocal of the sum of aerodynamic resistance, quasi-laminar sublayer resistance, and surface resistance. Over the land, surface resistance,

the major component of $v_d$, is classified into stomatal-mesophyll and several other resistance terms. Surface resistance is

usually strongly affected by its stomatal-mesophyll resistance term which in the Wesely scheme is expressed as seasonal- and LULC-dependent constants, which are subject to large uncertainty, being adjusted by surface temperature and radiation. This contrasts with some other approaches which also account for the influences of SM, vapor pressure deficit (VPD) and vegetation density, or couple stomatal resistance with photosynthesis. For calculating the other surface resistance terms, prescribed seasonal- and LULC-dependent constants are used in the Wesely scheme, adjusted by environmental variables including surface wetness, radiation and temperature, whereas in other existing schemes, impacts of friction velocity and vegetation density are also considered (e.g., Charusombat et al., 2010; Park et al., 2014; Val Martin et al., 2014; Wu et al., 2018; Mills et al., 2018b; Anav et al., 2018; Wong et al., 2019; Clifton et al., 2020, and the references therein). Aerodynamic resistance and quasi-laminar resistance are both sensitive to surface properties such as surface roughness.

This paper also briefly discusses in Section 3.4 some results from two WRF-Chem simulations (i.e., "SEACf" and "SEACa" in Table 1) during the 2013 Studies of Emissions and Atmospheric Composition, Clouds and Climate Coupling by Regional Surveys (SEAC[4]RS, Toon et al., 2016, https://espo.nasa.gov/home/seac4rs/content/SEAC4RS) campaign. SEAC[4]RS studies the attribution and quantification of pollutants and their distributions as a result of deep convection. These simulations were conducted on a 27 vertical layer, 25 km×25 km (99×67 grids) horizontal resolution Lambert conformal grid also centered at 33.5°N/87.5°W. Their LSM and inputs, WRF physics and chemistry configurations were the same as those used in the 12 km cases described above. In "SEACa", we assimilated successfully-retrieved, daily SM from version 04.5 of the European Space Agency Climate Change Initiative project (ESA CCI) SM product (Gruber et al., 2019), developed on a 0.25°×0.25° horizontal resolution grid based on measurements from passive satellite sensors. The assimilated CCI SM data were re-projected to the model grid and bias-corrected based on the climatology of Noah and CCI SM during August of 1999-2018. These simulations were evaluated with SEAC[4]RS aircraft chemical observations, which were richer than those collected during ACT-America in terms of the diversity of measured reactive chemical compounds (Section 2.2.1). Such comparisons help evaluate the emissions of $O_3$ precursors from various (e.g., NEI 2014 anthropogenic, lightning, and biogenic) sources as well as how the model representation of land-atmosphere interactions affects such emission assessments.

The model horizontal resolutions of 12 km and 25 km were set to be close to the assimilated satellite SM products to minimize the horizontal representation errors. At these resolutions, land surface heterogeneity and fine-scale processes (e.g., cloud formation and turbulent mixing) may not be realistically represented. Cloud-top-height-based lightning emissions and SM-precipitation feedbacks can be highly dependent on convective parameterizations (e.g., Hohenegger et al., 2009; Wong et al., 2013; Taylor et al., 2013). Addressing shortcomings of convective parameterizations in simulations at these scales is still in strong need. Performing convection-permitting simulations with assimilation of downscaled microwave SM or/and high-resolution thermal infrared based SM (e.g., 2-8 km from the Geostationary Operational Environmental Satellite) for cloudless conditions should also be experimented in the future.

## 2.2 Evaluation datasets

### 2.2.1 Aircraft in-situ measurements during ACT-America and SEAC$^4$RS

During the 2016 ACT-America deployment, the NASA B-200 aircraft took meteorological and trace gas measurements in the southeastern US from the surface to ~300 hPa on nine days. Different line colors in Figure 1d denote individual flight paths during this period. These flights were conducted under different weather conditions during the daytime (i.e., within 14-23 UTC, local time+6), with durations of 4-9 hours (https://www-air.larc.nasa.gov/missions/ACT-America/reports.2019/index.html). Flights on 16, 20, 21 of August 2016 sampled the air under stormy weather conditions, whereas the other flights were conducted under fair weather conditions. We used meteorological as well as collocated $O_3$ and CO measurements collected on the B-200 to evaluate our WRF-Chem simulations. The $O_3$ mixing ratio measurements using the differential ultraviolet absorption has a 5 ppbv uncertainty (Bertschi and Jaffe, 2005), and CO mixing ratio was measured with an uncertainty of 10 ppbv, using a Picarro analyzer which is based on wavelength-scanned cavity ring down spectroscopy (Karion et al., 2013). We used the weather and trace gas observations averaged in 1-minute intervals (version R1, released in November 2020) for model evaluation, as they represent atmospheric conditions on comparable spatial scales to the model. Ozone and CO measurements with $O_3/CO > 1.25$ mole mole$^{-1}$ (Travis et al., 2016) are assumed to be influenced by fresh stratospheric intrusions and were excluded in our analysis. This approach, however, was rather arbitrary and may not have excluded air that had an aged stratospheric origin or mixtures of air with different origins.

Aircraft (NASA DC-8, doi:10.5067/Aircraft/SEAC4RS/Aerosol-TraceGas-Cloud) in-situ measurements of CO, $NO_2$ and formaldehyde (HCHO) from the surface to ~200 hPa during six SEAC$^4$RS daytime (i.e., within 13-23 UTC, local time+6), 8-10-hour science flights in August 2013 were compared with our WRF-Chem simulations. The CO mixing ratio was measured using the tunable diode laser spectroscopy technique, with an uncertainty of 5% or 5 ppbv. The $NO_2$ measurements were made by two teams, based on thermal dissociation laser induced fluorescence and chemiluminescence methods, with the uncertainty of ±5% and (0.030 ppbv+7%), respectively. Two other teams took the HCHO measurements, using a compact atmospheric multispecies spectrometer and the laser-induced fluorescence technique, with the uncertainty of ±4% and (0.010 ppbv±10%), respectively. Aircraft data averaged in 1-minute intervals (version R7, released in November 2018) were used, with the biomass burning affected samples (acetonitrile >0.2 ppbv) and CO from fresh-stratospheric-intrusion-affected air ($O_3/CO > 1.25$ mole mole$^{-1}$) excluded.

### 2.2.2 Ground-based measurements

WRF-Chem results were evaluated by various surface meteorological and chemical observations. These include: 1) SM at ~5 cm and ~10 cm below the surface, measured at various sites within the Soil Climate Analysis Network (SCAN), which were downloaded from the International Soil Moisture Network (Dorigo et al., 2011) and screened by quality flags; 2) surface air temperature (T2), relative humidity (RH, derived from the original dew point and air temperature data), and wind speed

(WS) from the NCEP Global Surface Observational Weather Data (doi: 10.5065/4F4P-E398); 3) half-hourly or hourly latent and sensible heat fluxes measured using the eddy covariance method at eight sites within the FLUXNET network. Latent and sensible heat fluxes from this network exhibited mean errors of -5.2% and -1.7%, respectively (Schmidt et al., 2012). We only analyzed the modeled energy fluxes at the sites where the model-based LULC classifications are realistic. A 0.5°×0.5°, daily FLUXCOM product was also utilized, which merges FLUXNET data with machine learning approaches, remote sensing and meteorological data. Over North America, it is estimated that latent and sensible heat fluxes from this FLUXCOM product are associated with ~12% and ~13% of uncertainty, respectively (Jung et al., 2019); and 4) hourly $O_3$ at the US EPA Air Quality System (AQS, mostly in urban/suburban regions) and the Clean Air Status and Trends Network (CASTNET, mostly in nonurban areas) sites. Hourly AQS and CASTNET $O_3$ are US sources of the Tropospheric Ozone Assessment Report database, the world's largest collection of surface $O_3$ data supporting analysis on $O_3$ distributions, temporal changes and impacts. Measurements of $NO_2$ and HCHO are also available at some of the AQS sites. It is highly possible that these measurements are biased due to the interferences of other chemical species and therefore they were not used in this work.

### 2.2.3 Precipitation products

The WRF-Chem precipitation fields were also qualitatively compared with two precipitation data products: 1) the 4 km, hourly NCEP Stage IV Quantitative Precipitation Estimates (Lin and Mitchell, 2005), which is a widely-used, national radar and rain gauge based analysis product mosaicked from 12 River Forecast Centers over the contiguous US, and its quality partially depends on the manual quality control done at the River Forecast Centers; and 2) the 0.1°×0.1°, half-hourly calibrated rainfall estimates from version 6B of the Integrated Multi-satellitE Retrievals for the Global Precipitation Measurement (GPM) constellation final run product (Huffman et al., 2019). Compared with single-platform based precipitation products, multisensor based precipitation datasets have reduced limitations and therefore have become popular in scientific applications. Nevertheless, these datasets may be associated with region-, season-, and rainfall-rate dependent uncertainties (e.g., Tan et al., 2016; Nelson et al., 2016, and the references therein).

### 3 Results and discussions

### 3.1 Overview of the synoptic and drought conditions during the study periods

In August 2016, several states in the southern US experienced moderately-to-extremely moist conditions according to major drought indexes such as the Palmer Hydrological Drought Index (Figure S2, left). These were largely due to the influences of passing cold fronts and tropical systems from the Gulf of Mexico (https://www.ncdc.noaa.gov/sotc/synoptic/201608). Temperatures were consequently lower than normal in these regions. Contrastingly, controlled by the Bermuda High, more frequent air stagnation, warmer-, and drier-than-normal conditions affected multiple Atlantic states. Opposite hydrological

anomalies were recorded during August 2016 and August 2013 for the southern Great Plain and Atlantic regions (Figure S2, left).

The anomalies in synoptic patterns and drought conditions in August of 2016 and 2013, as well as the day-to-day weather changes, can partially explain the regional $O_3$ variability in the southeastern US. Based on the pressure gradients along the western edges of the Bermuda High (Zhu and Liang, 2012; Shen et al., 2015), the influences of the Bermuda High on southeastern US surface $O_3$ enhancements may be stronger in August 2016 than in August 2013 (Figure S2, middle). Lightning intensities and emissions respond to climate change (Romps et al., 2014; Murray, 2016; Finney et al., 2018), therefore affecting the probability of fires ignited by lightning. Based on satellite detections which are subject to cloud contamination, fire activities associated with emissions of heat and $O_3$ related pollutants were stronger in drier regions in the southern US in August of 2016 and 2013. The variable synoptic and drought conditions also controlled biogenic VOC and soil NO emissions as well as $O_3$-related chemical reaction and deposition rates, and the resulting impacts on $O_3$ depended on the changing anthropogenic $NO_x$ emissions (Hudman et al., 2010; Hogrefe et al., 2011; Coates et al., 2016; Lin et al., 2017). In the upper troposphere, troughs bumping into the anticyclone above the southeastern US in August 2016 helped shape the pollution outflows differently than in August 2013 when the North American monsoon anticyclone was built over the southwestern US and the central-eastern US was controlled by a strong cool trough (Figure S2, right).

Studies have shown that the variations in land-atmosphere coupling strength are connected with SM interannual variability and the local spatiotemporal evolution of hydrologic regime (e.g., Guo and Dirmeyer, 2013; Tuttle and Salvucci, 2016). Therefore, over the southern Great Plain and Atlantic regions, SM-atmosphere coupling strengths in August 2016 and August 2013 may have diverged from the climatology in opposite directions. For example, in August 2016, the overall potential impacts of SM on surface water/energy fluxes and atmospheric states may be higher than normal over the Atlantic regions whereas below the average in the southern Great Plain. In August 2013, the land-atmosphere coupling may be stronger than normal and abnormally weak over the southern Great Plain and the Atlantic regions, respectively.

### 3.2 Soil moisture, weather states and energy fluxes during ACT-America

Land and surface weather states as well as energy fluxes from the WRF-Chem base simulation, together with the SMAP DA impacts on these variables, are illustrated in Figure 2 (for SM), Figures 3-4 (for T2, RH, WS, and PBL height, PBLH), Figure 5 (for precipitation), Figures 6, S3 and S4 (for energy fluxes and their partitioning) for the 16-28 August 2016 period.

### 3.2.1 Observed and modeled SM and weather conditions

The highest daytime (13-24 UTC, local times+5 or +6) average T2 were observed in several states in the Atlantic region that were undergoing drought conditions (Figure S2, left; Figure 3b), The daily T2 maxima occurred during noon-early afternoon in most places, consistent with the findings from Huang et al. (2016). The Lower Mississippi River regions were influenced

by high humidity (Figure 3j). Under the influence of the Bermuda High, surface winds were overall mild to the east of Texas. Strongest rainfall affected Texas, Arkansas, Kentucky, Tennessee, and near the border of Kansas and Missouri (Figure 5a-b), which belonged to the wetter-than-normal regions according to August 2016 drought indexes. Rainfall in most areas peaked in the late afternoon or evening after the times of peak T2 (Figure 5e-f). The observed diurnal cycles of rainfall

and T2 indicate that, for the study area/period, convection was mainly due to the thermodynamic response to surface temperature. However, land-sea interactions, fronts, topography, as well as aerosol loadings may also have come into play.

The dry and wet anomalies in the southeastern US based on the modeled SM (Figure 2a) are shown to be consistent with ground-based SM measurements (e.g., Figure 2c), as well as weekly (not shown in figures) and monthly drought indexes (e.g., Figure S2, left). The modeled SM values in various soil layers are near the model-based soil wilting points and field

capacities (Figure S1, middle and lower) over drought-influenced and wetter-than-normal regions, respectively. The WRF-Chem base simulation overall captured the observed patterns of T2, RH, and WS across the domain, with its daytime PBLH spatially correlated with the T2 patterns (Table 2 and Figures 3a-b;d;i-j;k-l). Referring to the Stage IV and GPM rainfall data, the WRF-Chem base case also overall fairly well reproduced the diurnal cycles of rainfall during the study period, but the rainfall "hotspots" simulated by the model appear west to those in the Stage IV and GPM products (Figure 5c). Dirmeyer

et al. (2012) found that models' rainfall performance more strongly depended on the distinctive treatment of the model physics than on the model resolution. Our WRF-Chem performance for rainfall diurnal cycle in this region is similar to previous convection-permitting WRF-Chem simulations (e.g., Barth et al., 2012). Additionally, the WRF-Chem predicted mean rainfall rates over low-precipitation regions (e.g., several Atlantic states) are higher than those based on the Stage IV and GPM rainfall products, which tend to overestimate precipitation at the low end (e.g., Nelson et al., 2016; Tan et al.,

2016). Such positive model biases for low-precipitation regions have also been reported in Barth et al. (2012).

### 3.2.2 SMAP DA impacts on SM, surface weather states and energy fluxes

Surface SM at the model initial times (i.e., 00 UTC each day) was broadly reduced by the SMAP DA, except parts of coastal Texas, Ohio and Florida (Figure 2d). Such changes in the modeled SM fields are consistent with the modeled daytime specific humidity (not shown in figures) and RH responses (Figure 3m). They are anti-correlated with the model responses in

its averaged daytime T2 and PBLH fields (Figure 3e;h) as well as their daily amplitudes (not shown in figures). The daytime T2 and RH responses to the SMAP DA are statistically significant in ~21% and ~65% of the overland model grids, respectively (i.e., $p<0.05$ based on the Student's t-tests, Figure 4a-b), with the most significant daytime-averaged responses of ~2 K and >10%, respectively, occurring in Missouri and Ohio, as well as several other states located within 33-40 °N and 90-100 °W. In places, the daily maxima of WRF-Chem T2 were delayed by an hour or two when the SMAP DA was enabled

(Figure 3g). The changes in WRF-Chem temperature gradients due to the SMAP DA led to slight WS enhancements over many of the model grids (Figure 3o). In contrast to the WRF-Chem T2 and RH responses, these WS changes are statistically insignificant (i.e., $p>0.05$ based on the Student's t-tests) in ~97% of the overland model grids (Figure 4c). On the 13-day

timescale, the SMAP DA had less discernable impacts on rainfall, consistent with the findings from Koster et al. (2010, 2011) and Huang et al. (2018). The SMAP DA impacts on mean rainfall rate and diurnal cycles show noisy patterns (Figure 5d;g;h), and positive and negative SM-precipitation relationships are both found. The spatial and temporal variability in these model sensitivities reflects the impacts of local hydrological regimes and their anomalies as well as moisture advection.

It is indicated by Figure 2b;e that, during the study period, the SMAP DA successfully reduced the discrepancies between SMAP and Noah-calculated surface SM across the model domain. The modeled surface SM was also cross-validated with ground-based SM measurements at dozens of SCAN sites, using the root-mean-square error (RMSE) metric. Figure 2f-g shows the results based on a comparison of the modeled surface SM with ~10 cm belowground SM measurements at these SCAN sites. This evaluation suggests that the Noah-based SM was more evidently improved by the SMAP DA at sparsely-vegetated regions, i.e., RMSE was reduced at almost all sites where green vegetation fraction ≤0.6. At dense-vegetation (i.e., green vegetation fraction >0.6) SCAN sites, over a half of which are located in cropland areas subject to the impacts of irrigation and other human activities, the SMAP DA did not prevalently decrease or increase the discrepancies between the modeled and measured SM. Similar findings were reached based on such a comparison of the modeled surface SM and ~5 cm belowground SM measurements at these SCAN sites. The overall T2, RH and WS performance of WRF-Chem was not prevalently improved or degraded due to the inclusion of the SMAP DA (e.g., Figure 3f;n;p, based on the RMSE metric): i.e., improvements on T2, RH, and WS occurred in 47%, 51% and 52% of the model grids where observations are available, and the domain-wide mean RMSE changes for T2, RH, and WS are ~0 K, -0.024%, and -0.005 ms$^{-1}$, respectively (Table 2). This finding for dense vegetation regions is qualitatively consistent with those in Huang et al. (2018) and Yin and Zhan (2018) which are based on RMSE and other evaluation metrics, and it may partially be attributed to SMAP retrieval quality and the land-atmosphere feedbacks represented in Noah. Additionally, as discussed in Huang et al. (2018), unrealistic model representations of terrain height can pose challenges for evaluating the modeled surface weather fields with ground-based observations. The 12 km model grid used in this work well represents terrain height (i.e., |model-actual|<15 m) at over 70% of the model grids that have collocated observations, but at some locations the discrepancies between the model and actual terrain height exceed 100 m. Furthermore, human activities such as irrigation can significantly modify water budget and land-atmosphere coupling strength over agricultural regions (e.g., Lu et al., 2017), but these processes were unaccounted for in the modeling system used. Observations from SMAP and other satellites are capable of detecting the signals of irrigation over the southeastern US (e.g., the circled regions in Figure 1c based on Ozdogan and Gutman (2008) and Zaussinger et al. (2019)) and other regions of the world. However, for locations where irrigation or/and other missing processes dominantly contributed to the systematic biases between the modeled and SMAP SM, the bias correction approach applied may have removed the information of these processes from the SMAP observations before the DA. As a result, the DA may not be effective at these locations. How irrigation patterns and scheduling, depending in part on the weather conditions, affected our WRF-Chem performance as well as the effectiveness of the SMAP bias correction and DA are worth further investigations. In places, the changes in WRF-Chem rainfall patterns due to the SMAP DA are within the discrepancies between the Stage

IV and GPM rainfall products. A better understanding of the uncertainty associated with these two used rainfall products can benefit the assessment of SM DA impacts on the model's precipitation performance.

The spatial patterns of evaporative fraction (defined as: latent heat/(latent heat+sensible heat)) follow those of SM and RH, with the maxima (>0.75) seen in the Lower Mississippi River region and smaller values (<0.65) in the dry Atlantic states and some parts of the southern Great Plains (Figure 6a-b). Note that the absolute latent and sensible heat fluxes can differ significantly at locations with similar evaporative fraction values (Figure S3). The WRF-Chem based evaporative fraction shows similar spatial gradients but is overall negatively biased (Figure 6c). The changes in WRF-Chem evaporative fraction due to the SMAP DA are spatially correlated with the surface moisture changes (Figure 2d;3m;6d). As a result, the model performance of evaporative fraction was only improved over some of the regions where it was increased by the SMAP DA. It is found that the SMAP DA impacts on model performance are not universally consistent for surface energy fluxes and land/atmosphere states. This can be explained by the fact that the modeling system used has shortcomings in representing SM-flux coupling and/or the relationships between moisture/heat fluxes and the atmospheric weather which need to be clearly identified and corrected. The most possible reasons causing such model behaviors include: 1) irrigation and other processes related to human activities were unaccounted for, and the surface exchange coefficient $C_H$, which is a critical parameter controlling energy transport from the land surface to the atmosphere, may not be realistically represented in Noah (details in Section S1); 2) the SMAP DA did not update the vegetation and surface albedo fields in Noah, which was unrealistic; and 3) soil parameters determined from soil texture types and a lookup table may be inaccurate in places. To confirm and address these limitations in the modeling/DA system used, and to identify other possible reasons, future efforts should be devoted to: applications using other LSMs (e.g., the Noah-Multiparameterization), up-to-date inputs and parameters (e.g., soil texture types and lookup tables), together with multivariate land DA; evaluation of additional water and energy flux variables such as runoff and radiation, the latter of which shows inconsiderable sensitivities to the SMAP DA (Figure S4); and utilization of alternative WRF inputs and physics configurations.

### 3.2.3 SMAP DA impacts on weather conditions at various altitudes

The WRF-Chem modeled weather states were also evaluated with ACT-America aircraft observations at various altitudes. Along the flight paths, the observed air temperature and water vapor mixing ratios decrease with altitude, which were fairly well captured by WRF-Chem (Figures 7a-b;e-f and 8a;c). The modeled air temperature and humidity as well as their responses to the SMAP DA vary in space and time. In general, these responses are particularly strong near the surface, where the majority of the samples were collected. Under stormy weather conditions on 16, 20, 21 of August 2016, the maximum changes in air temperature and humidity in the free troposphere exceed 2.3 K and 2 gkg$^{-1}$, respectively (Figure 7c;g). Corresponding to these changes, the SMAP DA modified the RMSEs of WRF-Chem air temperature and/or water vapor by over 5% for several individual flights and overall reduced the RMSEs of these model variables by ~0.7% and ~2.3%, respectively (Figures 8b). The most significant improvements in the modeled weather states occurred at >=800 hPa, where

the maximum improvements in air temperature and water vapor exceed 2.6 K and 2 $gkg^{-1}$, respectively, and their RMSEs were both reduced by ~2.7% (Figures 7d;h and 8d).

## 3.3 Ozone and its responses to the SMAP DA during ACT-America

### 3.3.1 Surface $O_3$

The changes in the above-discussed meteorological variables (e.g., air temperature, humidity, WS, PBLH) due to the SMAP DA alter various atmospheric processes which can have mixed impacts on surface $O_3$ concentrations. For example, warmer environments promote biogenic VOC and soil NO emissions as well as accelerate chemical reactions (e.g., many oxidation processes, thermal decomposition of peroxyacetyl nitrate). These will be discussed in detail in the following paragraphs referring to Figures 9 and S5. Faster winds and thickened PBL dilute air pollutants including $O_3$ and its precursors, and therefore reduce $O_3$ destruction via titration (i.e., $O_3+NO \rightarrow O_2+NO_2$) as well as photochemical production of $O_3$. The changes in wind vectors affect pollutants' concentrations in downwind regions. Water vapor mixing ratios perturb $O_3$ photochemical production and loss via affecting the $HO_x$ cycle. Their impacts on $O_3$ levels depend on the chemical environments of the areas of interest, i.e., in general, reduced specific humidity slightly enhances $O_3$ except in some polluted regions. Also, higher RH often has relevance with cloud abundance and solar radiation and therefore slow down the photochemical processes (Camalier et al., 2007). Additionally, chemical loss via stomatal uptake may be slower under lower-SM/humidity, higher-temperature conditions, and nonstomatal uptake also varies with meteorology. These processes, however, may not all be realistically represented by the Wesely dry deposition scheme (Sections 2.1 and S2; Figures S1 and S7) used in this study.

Figure 10a-b compare the observed and WRF-Chem base case daytime surface $O_3$ during 16-28 August 2016, and the SMAP DA impacts on daytime surface $O_3$ are shown in Figure 10c. Low-to-moderate $O_3$ pollution levels are seen over most areas within the model domain, except the Atlantic states due to the influences of frequent air stagnation, warm and dry conditions. Period-mean daytime surface $O_3$ responses to the SMAP DA are overall slightly positive, but exceed or closely approach 2 ppbv in some places in Missouri, Illinois, and Indiana, and the strongest decreases in the period-mean daytime surface $O_3$ occurred in Ohio (i.e., by >2 ppbv). The averaged $O_3$ changes show strong spatial correlations (with correlation coefficient $r$ values of ~0.8) with those of T2 and PBLH (Figure 3e;h), which are anti-correlated with the surface humidity responses (Figures 2d and 3m). On most of the days during 16-28 August 2016, the maximum impacts of SMAP DA on daily daytime surface $O_3$ exceed 4 ppbv, and the $O_3$ sensitivities are moderately correlated with the daytime T2 changes (Figure 11a, with $r$ values within 0.4-0.7). The period-mean WRF-Chem surface MDA8 and its response to the SMAP DA (Figure 12a-b) show similar spatial patterns to those of the modeled surface daytime $O_3$, but are of higher variability.

The overall enhanced biogenic emissions of VOCs and soil NO (Figures 9a-b;e-f and S5, first two rows) belong to the major causes of the changes in the surface daytime-average and MDA8 $O_3$ described above. The SMAP DA impacts on MEGAN biogenic emissions were largely due to its impact on T2 (Figure 3e). This is because the modeled photosynthetically active radiation (PAR), which is another variable critical to estimating biogenic emissions of some species, shows only <<10% of responses to the SMAP DA in most places (Figure S4, lower), and based on previous MEGAN emission-PAR sensitivities analysis (e.g., Figure 2 in Guenther et al., 2012), it is estimated that these changes in modeled PAR have caused negligible impacts on the modeled biogenic emissions. MEGAN biogenic emissions were most strongly modified over the regions with elevated emissions: i.e., by >20% over the Missouri Ozarks for isoprene and by >10% over agricultural land for soil NO, where emission factors at standard conditions are high and the DA-induced T2 changes are strong and statistically significant. Over the Missouri Ozarks, the >20% isoprene emission changes corresponding to the ~2 K T2 changes are consistent with the previously-reported isoprene emission sensitivities to surface air temperature (e.g., Huang et al., 2017b, and the references therein). MEGAN's limitations in representing biogenic VOC emission responses to drought may have had minor impacts on most of the high-biogenic-emission regions which were not affected by drought during this period. For certain parts of the Atlantic states that were in the early-middle phases of drought in August 2016 referring to drought indexes from July-October 2016 (not shown in figures), while it is highly likely that the reserved carbon resources were still available and leaf temperature still controlled the VOC emissions, the lack of SM-dependency in MEGAN VOC emission calculations may have introduced uncertainty to the results from both the base and the "assim" cases. However, as the SMAP DA only mildly affected SM and T2 over these regions (Figures 2d and 3e), we do not anticipate that biogenic VOC emissions would be changed significantly there by the SMAP DA even if their dependency on SM was realistically included in MEGAN. Also, note that for this case satellite-based leaf area index data were used in MEGAN BVOC emission calculations. Although satellite-based leaf area index data may be more accurate than those calculated by dynamic vegetation models, they are less temporally-variable than the reality, and the SMAP DA did not adjust this critical MEGAN input. These also limited the responses of MEGAN-calculated VOC emissions (and thus $O_3$-related chemical fields) to the DA. Uncertainty in the modeled soil NO emissions and their responses to the SMAP DA may be larger over high-temperature cropland regions. This needs further investigations accounting for the influences of SM, which is controlled by both precipitation and human activities such as irrigation, as well as the fertilization conditions.

The overall accelerated chemical reactions, including those strongly controlling the lifetime of peroxyacetyl nitrate, are also highly responsible for the above-mentioned changes in surface daytime-average and MDA8 $O_3$. For example, in broad regions north of 33 °N, the modeled daytime-mean surface peroxyacetyl nitrate concentrations show 10-20% responses to the SMAP DA, and these responses are mostly in the opposite directions of the T2 and surface $O_3$ changes (Figures 9c;g). This reflects that the increased (decreased) temperatures sped up (slowed down) the decomposition of peroxyacetyl nitrate which formed the $O_3$-production-related peroxyacetyl radical and $NO_2$.

The $v_d$ of $O_3$ and its related chemical species also responded to the SMAP DA, with the changes in $v_d$ of $O_3$ (written as $v_{d[O3]}$ thereafter) estimated to be the most important to the modeled $O_3$ concentrations according to previous studies (e.g., Baublitz et al., 2020). The modeled daytime $v_{d[O3]}$ responses to the SMAP DA, as well as those in the major, stomata-related term of $v_{d[O3]}$, are found to be anti-correlated with those in surface temperature (Figures 9d;h, S5, lower, and S6). Although surface radiation also adjusts some $v_d$ terms, in this work it insignificantly responded to the SMAP DA (Figure S4, upper) and therefore contributed much less importantly than surface temperature to the modeled $v_{d[O3]}$ changes. The responses of $v_{d[O3]}$ are within $\pm0.02$ cms$^{-1}$ in >70% of the model grids but are outside of $\pm0.05$ cms$^{-1}$ in some high $v_{d[O3]}$ regions such as Missouri and Ohio (i.e., base case $v_{d[O3]}>0.7$ cms$^{-1}$) where they were highly responsible for the surface $O_3$ changes. Note that these $v_{d[O3]}$ results are based on the Wesely scheme in which the SM and VPD influences on stomatal resistance are omitted. If SM and VPD limitation factors (details in the captions of Figures S1 and S7) were included in the calculations of stomatal resistance, the modeled $v_d$ in both the base and the "assim" cases would become smaller, especially over dry environments, and the SMAP DA may result in more intense relative changes in the modeled $v_d$. Including such SM and VPD limitation factors in $v_d$ calculations, however, would not necessarily improve the modeled $v_d$ in part due to the uncertainty in the model's LULC input and the prescribed seasonal- and LULC-dependent constants in the Wesely scheme used. Future efforts need to be devoted to quantifying how the SMAP DA influences $v_d$ calculations in a modeling/DA system with dynamic vegetation and the $v_d$ parameterizations are coupled with photosynthesis and vegetation.

The SMAP DA improved surface MDA8 at 42% and 51% of the model grids where AQS and/or CASTNET observations are available, respectively. It increased the domain-wide mean MDA8 RMSEs by 0.057 ppbv and 0.007 ppbv referring to the gridded AQS and CASTNET $O_3$ observations, respectively. The MDA8 RMSEs were shown increased in some of the areas (e.g., a few sites in Ohio) where the modeled SM, surface weather fields and energy fluxes were improved by the SMAP DA (Figures 2f, 3f;n, 6, and 12e). As summarized in Table 3, after enabling the SMAP DA, the number of grids with $O_3$ exceedance false alarms (i.e., WRF-Chem MDA8 $O_3>70$ ppbv but the observed MDA8 $O_3<=70$ ppbv) remained the same, except that this number dropped on 26 August and increased on 18 August. The less desirable $O_3$ performance changes in response to the SMAP DA than those in the weather fields can be explained by the fact that many other factors, such as the quality of the anthropogenic emission input of WRF-Chem, also affected the model's surface $O_3$ performance. Figures 12c;f and 11b show that using NEI 2016 beta anthropogenic emissions instead of the outdated NEI 2014 resulted in notable reductions in surface daytime-average and MDA8 $O_3$ across the model domain. These reductions lowered the modeled surface $O_3$ biases by up to ~4 ppbv and reduced the number of grids with $O_3$ exceedance false alarms on 7 out of the 13 days (Table 3). Improving the modeled weather fields via the SMAP DA would more clearly improve the model's $O_3$ performance if the uncertainty of NEI 2016 beta and other inputs as well as the model parameterizations (e.g., chemical mechanism, natural emission, photolysis and deposition schemes) is reduced.

It is noticed that daytime surface $O_3$ fields from the global CAMS and AM4 modeling systems are overall higher than those simulated by WRF-Chem (Figure 10b;d;e). One of the reasons is that stratosphere-troposphere exchanges are better represented in these two global models. According to AM4's stratospheric tracer, during the study period, the stratospheric $O_3$ influences on daytime surface $O_3$ range from <2 ppbv in the southern Great Plains (storm-affected regions) to 6-7 ppbv around Kansas and the Atlantic Ocean. Note that although AM4 provides a broad overview of the areas strongly impacted by stratospheric air, fine-scale features associated with stratospheric intrusions may be missing from this coarse-resolution simulation (Lin et al., 2012; Ott et al., 2016). Figure S7 (middle) indicates that the WRF-Chem modeling system used is capable of reproducing the downward and upward movements of pollutants: i.e., positive vertical wind speeds are shown over storm-active regions and negative vertical wind speeds over many regions that were strongly affected by stratospheric $O_3$. However, as this modeling system has only tropospheric chemistry, the influences of stratospheric chemical compounds are represented only through the model's chemical LBCs. This representation may be improved by adding accurate, time-varying chemical upper boundary conditions, e.g., downscaled from a fine-resolution (e.g., with horizontal spacing <50 km), well-performed global model simulation. Such an update, however, is expected to increase the modeled surface $O_3$ (e.g., Figure 3 in Huang et al., 2013, based on a different regional air quality model). For regions where modeled surface $O_3$ is already positively biased, stronger efforts to address other sources of model errors would be needed to achieve desirable surface $O_3$ performance.

### 3.3.2 Ozone at various altitudes

The SMAP DA impacts on WRF-Chem modeled chemical fields are also investigated at a wide range of altitudes. Figure 7i-p compare the observed and WRF-Chem base case CO and $O_3$ concentrations along nine ACT-America flights in August 2016, as well as the SMAP DA impacts on WRF-Chem results at these sampling locations. The observed and modeled CO vertical profiles show strong day-by-day variability, with near-surface concentrations ranging from 60 to 170 ppbv and elevated concentrations aloft (>90 ppbv at <600 hPa) occurring on 16, 20, 21 of August when aircraft measurements were taken under stormy weather conditions. In general, the observed and modeled $O_3$ increase with altitude. WRF-Chem fairly well captured the magnitudes of the near-surface $O_3$ concentrations but underpredicted $O_3$ in the free troposphere. Overall, the modeled trace gas concentrations reacted to the SMAP DA most strongly near the surface. Under stormy weather conditions, the maximum changes in modeled CO and $O_3$ approach 20 ppbv and 10 ppbv, respectively, corresponding to improved model performance at these locations (Figure 7k-l;o-p). The SMAP DA impacts on modeled CO and $O_3$ RMSEs are overall close to neutral (|ΔRMSE|<0.5%) but over 2% during selected flights (Figures 8b). Similar to the evaluation results for surface weather and $O_3$ fields, the $O_3$ performance changes by the SMAP DA are less desirable than those in the weather fields.

To help better understand SM controls on upper tropospheric $O_3$ chemistry, Figures 13d-i and S7 (lower) show the period-mean (16-28 August 2016) daytime $O_3$, CO, $NO_2$ and lightning $NO_x$ tracer results at ~400 hPa from the WRF-Chem base

simulation, as well as the SMAP DA impacts on these model fields. The daily daytime $O_3$ responses to the SMAP DA at ~400 hPa are presented in Figure 11c. Elevated WRF-Chem $O_3$ concentrations (>70 ppbv) are seen near the center of the upper-tropospheric anticyclone (Figure S2, right), which circulated the lifted pollutants and promoted in-situ chemical production. The SMAP DA modified the period-mean daytime $O_3$ by up to 1-1.5 ppbv, and its impacts on daytime $O_3$ on individual days during the study period occasionally exceed 10 ppbv, which is larger than its maximum impact on the daily

daytime surface $O_3$ (Figure 11a;c). As indicated by the modeled CO as well as $NO_2$ and lightning $NO_x$ tracer responses to the SMAP DA, the $O_3$ distributions in the upper troposphere and their responses to the SMAP DA are partially controlled by atmospheric transport and rapid in-situ chemical production of $O_3$ from lightning NO and other emissions, both of which are sensitive to SM. CO is used here primarily as a tracer of transport, but note that lightning and other emissions can modify CO lifetimes.

Similar to the $O_3$ conditions at the surface, at ~400 hPa, WRF-Chem daytime $O_3$ concentrations are lower than the global CAMS and AM4 results (Figure 13a-b) as well as the ACT-America aircraft measurements (Figure 7m-n), by up to tens of ppbv. The AM4 stratospheric tracer suggests 5-17 ppbv of stratospheric influences on the period-mean $O_3$ at these altitudes (Figure 13c), which again helps identify the shortcoming of WRF-Chem in representing stratosphere-troposphere exchanges. Applying accurate, time-varying chemical upper boundary conditions in future works can help better assess the SMAP DA

impact on $O_3$ performance in the upper troposphere and improve the understanding of upper tropospheric chemistry.

To help interpret the SMAP DA impacts on various atmospheric processes such as vertical transport and lightning associated with convection and other phenomena, model results from the base and the "minus001" cases during two ACT-America flights were compared (Figure S8). In the afternoon of 20 August 2016, the B-200 flew at <500 hPa over cold regions in Oklahoma and Arkansas affected by convection with a cold front involved. On 27 August 2016 when most southeastern US

regions were experiencing fair and warm weather, some of the B-200 measurements were collected at <400 hPa over the southern Mississippi influenced by deep convection. The WRF-Chem modeled CO concentrations in the free troposphere above the regions affected by the cold front and/or convection are shown strongly sensitive to surface SM, and AM4 stratospheric $O_3$ tracer output suggests enhanced stratospheric influences near the cold front and/or convection-affected locations. While this sensitivity analysis based on a constant surface SM perturbation helped confirm the SM impacts on

atmospheric weather and chemistry, it is important to note that in reality the SM-atmosphere feedbacks are controlled by the magnitude and spatial heterogeneity of SM which were both adjusted by the SMAP DA. Figure 7k-l shows that the SMAP DA improved the WRF-Chem CO concentrations in the upper troposphere during both of these flights.

It is also noticed that the daytime $O_3$ changes related to the anthropogenic emission update from NEI 2014 to NEI 2016 beta (<20% of change for most species as introduced in Section 2.1) have comparable magnitudes with those due to the SMAP

DA in the upper troposphere. For example, at ~400 hPa, those changes are mostly within ±10 ppbv and ±1.5 ppbv at daily and 13-day timescales, respectively (Figures 11c-d;13g and S9, upper). This suggests that the SMAP DA and the US EPA

estimated anthropogenic emission change from 2014 to 2016 over the southeastern US could have similar levels of impacts on modeled $O_3$ export from this region. The magnitudes of WRF-Chem upper-tropospheric $O_3$ sensitivities to anthropogenic emissions and SM are close to those based on archived global model sensitivity simulations for August 2010 which quantify monthly $O_3$ responses to a constant 20% reduction in North American anthropogenic emissions (i.e., 0.7-1.5 ppbv, Figure S9, lower). Those global model simulations also estimated that this 20% emission reduction in North America affected $O_3$ in other regions of the world: e.g., ~400 hPa and surface $O_3$ in Europe decreased by 0.4-0.7 ppbv and 0.1-0.5 ppbv, respectively (Figure S9, lower-middle). Our WRF-Chem results, together with the findings from these past global model experiments, suggest that SM plays an important role in quantifying air pollutants' source-receptor relationships between the US and its downwind regions. It also emphasizes that using outdated anthropogenic emissions in WRF-Chem would lead to inaccurate assessments of the SMAP DA impacts on the model performance of $O_3$ and other air pollutants over a broad region.

### 3.4 Evaluation of NEI 2014 using WRF-Chem simulations and SEAC[4]RS observations

We compared CO, $NO_2$, and HCHO from two 25 km WRF-Chem simulations (i.e., the "SEACf" and "SEACa" cases, Table 1) with aircraft observations during six SEAC[4]RS flights in August 2013 (Figure S10). Such comparisons help evaluate the emissions of $O_3$ precursors from various (e.g., NEI 2014 anthropogenic, lightning and biogenic) sources as well as how the model representation of land-atmosphere interactions can affect such emission assessments. It is shown that in case "SEACf", WRF-Chem reproduced the overall vertical gradients of the observed chemicals, except that at this resolution it had difficulty in capturing urban plumes (e.g., for where the observed $NO_2$ >4 ppbv). This suggests that emissions of major $O_3$ precursors are moderately well represented in the WRF-Chem system used. The strongest improvements in modeled CO, $NO_2$, and HCHO by assimilating the CCI SM are ~12 ppbv, ~0.6 ppbv, and ~1.2 ppbv, respectively, all occurring near the surface (>700 hPa). In the upper troposphere, the SM DA enhanced the modeled CO by up to ~6 ppbv (at ~200 hPa) and reduced the modeled $NO_2$ by up to ~0.5 ppbv (at ~400 hPa). These changes led to better model agreements with the observations, indicating that assimilating the CCI SM likely improved the model treatment of lightning production and convective transport. As the SM DA modified the mismatches between the modeled and the observed trace gas concentrations, it is suggested that accurate representations of land-atmosphere interactions can benefit more rigorous evaluation and improvement of emissions using observations. Additionally, aircraft observations show robustness in aiding the evaluation of the emissions of $O_3$ precursors from various sources, and therefore continuing to make rich and detailed observations like those would be helpful for evaluating and improving newer/future versions of emission estimates as well as the model representations of land-atmosphere interactions.

### 4 Summary and suggestions on future directions

This study focused on evaluating SMAP SM DA impacts on coupled WRF-Chem weather and air quality modeling over the southeastern US during the ACT-America campaign in August 2016. The impacts of SMAP DA on WRF-Chem modeled

daytime RH as well as evaporative fraction were qualitatively consistent with the changes in the model's initial SM states, which were anti-correlated with the modeled daytime surface T2 and PBLH changes. The DA impacts on the model performance of SM, weather states and energy fluxes showed strong spatiotemporal variability. Many factors may have impacted the effectiveness of the DA, including missing processes such as water use from human activities (e.g., irrigation), as well as dense vegetation and complex terrain as also discussed in detail in our previous SMAP DA study. Referring to the gridded NCEP surface observations, the domain-wide mean RMSEs of modeled T2, RH, and WS were changed by the DA by ~0 K, -0.024%, and -0.005 ms$^{-1}$, respectively. Referring to ACT-America aircraft observations on nine flight days, the DA reduced the RMSEs of WRF-Chem air temperature and water vapor by ~0.7% and ~2.3%, respectively. The most significant improvements in the modeled air temperature and humidity occurred at >=800 hPa, where their RMSEs were both reduced by ~2.7%. The overall DA impact on the modeled rainfall was less discernable, within the discrepancies between two rainfall evaluation products in places. The DA impacts on model performance were not consistent for energy flux partitioning and land/atmosphere states everywhere, suggesting that the modeling system used had shortcomings in representing SM-flux coupling and/or the relationships between moisture/heat fluxes and the atmospheric weather which need to be more clearly identified and corrected. Future efforts should focus on: 1) applications using other LSMs, up-to-date inputs and parameters, along with multivariate land DA; 2) evaluation of additional water and energy flux variables (e.g., runoff, radiation); and 3) utilization of alternative LIS/WRF configurations, including adding irrigation processes to the modeling system and performing convection-permitting simulations with the assimilation of various kinds of high-resolution land products. Additionally, improving bias correction methods (e.g., also matching higher-order moments of the LSM and satellite SM climatology) and practicing the assimilation of SMAP Level 1 brightness temperature alone or in combination with atmospheric observations will be needed.

The SMAP DA impact on WRF-Chem surface daytime-average and MDA8 $O_3$ were strongly correlated with the changes in daytime T2 and PBLH, which were anti-correlated with the daytime surface humidity changes. The DA-induced surface $O_3$ changes can largely be explained by the temperature-driven changes in biogenic emissions of VOCs and soil NO, chemical reaction rates, as well as dry deposition velocities. The SMAP DA impacts on WRF-Chem modeled $O_3$ along the ACT-America flight paths were particularly strong (i.e., approaching 10 ppbv at some >=800 hPa locations) under stormy weather conditions. The WRF-Chem (near-)surface $O_3$ performance change in response to the DA was overall less desirable than those in the weather fields, e.g., referring to gridded AQS and CASTNET $O_3$ observations, the domain-wide mean MDA8 RMSEs increased by 0.057 ppbv and 0.007 ppbv, respectively. This was in part because many other factors also affected the model's surface $O_3$ performance, such as shortcomings in model parameterizations (e.g., chemical mechanism, natural emission, photolysis and deposition schemes) and the model representations of anthropogenic emissions and stratosphere-troposphere exchanges.

We showed that at ~400 hPa, elevated $O_3$ concentrations were modeled near the center of the upper tropospheric anticyclone. The modeled $O_3$ was negatively biased, mainly resulting from the poor representation of stratosphere-troposphere exchanges by WRF-Chem. The impact of SMAP DA on upper tropospheric $O_3$ was partially via altering the transport of $O_3$ and its precursors from other places as well as in-situ chemical production of $O_3$ from lightning NO and other emissions (including $O_3$ precursors transported from elsewhere). Case studies of convection and/or cold front-related events suggested that the DA improved the model treatment of convective transport and/or lightning production, which strengthened and extended the findings in Huang et al. (2018). We also presented that the impacts of DA and an emission update from NEI 2014 to NEI 2016 beta on WRF-Chem upper tropospheric $O_3$ had comparable magnitudes. As reducing North American anthropogenic emissions would benefit the mitigation of $O_3$ pollution in its downwind regions, our analysis highlighted the important role of SM in quantifying air pollutants' source-receptor relationships between the US and its downwind areas. It also emphasized that using up-to-date anthropogenic emissions in WRF-Chem would be necessary for accurately assessing SM DA impacts on the model performance of $O_3$ and other air pollutants over a broad region. Continuing to improve NEI 2016 beta and any newer versions of emission estimates, as well as the parameterizations and other inputs of the models, is strongly encouraged. Such efforts can benefit from rich, detailed, high-accuracy observations, such as those taken during airborne field campaigns.

This study is a critical first step towards using satellite SM products to help improve the simulated weather and chemistry fields in models that are widely-used for air quality research and forecasting, as well as policy-relevant assessments. It was demonstrated that, via changing the model's weather fields that drove its chemistry calculations online, the SM DA influenced various $O_3$-related processes, $O_3$ concentrations and exceedances modeled by WRF-Chem. In some locations/times, these influences were significant and resulted in improved model performance. To further improve the modeled chemical fields via applying the SM DA at various scales, it is not only important to improve the model representations of anthropogenic emissions and trans-boundary transport, but also to address shortcomings in model parameterizations, e.g., to realistically reflect the impacts of water availability on biogenic emissions and dry deposition, and for longer simulations, to include $O_3$ damage to vegetation (e.g., Hudman et al., 2012; Val Martin et al., 2014; Sadiq et al., 2017; Jiang et al., 2018; Clifton et al., 2020). Using dynamic vegetation models (available in the Noah-Multiparameterization LSM) along with additional process-based (e.g., chemical fluxes, stomatal behaviors) measurements and laboratory experiments would be necessary for improving some of these parameterizations, and these will be experimented in a follow-up study. Community efforts such as the ongoing Air Quality Model Evaluation International Initiative Phase 4 experiment (https://aqmeii.jrc.ec.europa.eu/phase4.html) would also be greatly beneficial. High-quality weather input is a requirement for rigorous evaluations of any set of these parameterizations.

**Code and data availability**

The standalone LIS is accessible at: https://lis.gsfc.nasa.gov. LIS/WRF-Chem coupling is facilitated in the NASA-Unified WRF system (https://nuwrf.gsfc.nasa.gov). The global C-IFS simulations for HTAP2 are available at the AeroCom database. Observations and observation-derived data products used in this work can be found at: https://nsidc.org/data/smap/smap-data.html; https://ismn.geo.tuwien.ac.at; https://www.esa-soilmoisture-cci.org; https://www-air.larc.nasa.gov/index.html;
https://www.epa.gov/aqs; https://www.epa.gov/castnet; https://rda.ucar.edu/datasets/ds461.0; https://fluxnet.fluxdata.org; http://www.fluxcom.org; https://www.emc.ncep.noaa.gov/mmb/ylin/pcpanl/stage4; and https://pmm.nasa.gov/data-access/downloads/gpm.

**Author contributions**

MH led the design and execution of the study as well as the paper writing. JHC, JPD, GRC, and KWB contributed to the
field campaign data collection and/or analysis. GRC, KWB, SVK and XZ contributed to the modeling and/or DA work. All authors helped finalize the paper.

**Competing interests**

The authors declare that they have no conflict of interest.

**Acknowledgements**

We thank the ACT-America flight, instrument and data management teams, and the ACT-America Principal Investigator, Kenneth Davis (Penn State), for designing and conducting the NASA B200 flights, as well as helping with the analysis. We also thank the SEAC$^4$RS instrument teams (PIs: Thomas Ryerson, NOAA/ESRL; Ronald Cohen, UC Berkeley; Alan Fried, CU-Boulder; Thomas Hanisco, NASA GSFC; Glenn Diskin, NASA LaRC; and Armin Wisthaler, University of Innsbruck) and FLUXNET PIs for sharing their measurements. The ECMWF CAMS and GFDL AM4 (contacts: Meiyun Lin, Princeton;
Alex Zhang, now at Penn State) modeling teams are acknowledged for generating the global model outputs used in this study. The global C-IFS simulations for HTAP2 were conducted by Johannes Flemming (ECMWF). Sophia Walther (MPI-BGC), Kazuyuki Miyazaki (JPL/Caltech), and Li Fang (UMD) provided datasets that are not directly related to this study but informative. NASA SUSMAP sponsorship for this work as well as NASA's high-end computing systems and services is acknowledged. The ACT-America project is a NASA Earth Venture Suborbital 2 project funded by NASA's Earth Science
Division (Grant NNX15AG76G to Penn State).

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

**Tables**

**Table 1: Summary of WRF-Chem simulations conducted in this study.**

| Case name | Horizontal/vertical resolutions | Analyzed period (field campaign) | Assimilated SM data (version; resolution) | Anthropogenic emission inputs for various chemical species |
|---|---|---|---|---|
| Base | | 16-28 August 2016 (ACT-America) | none | NEI 2016 beta |
| Assim | | | SMAP enhanced passive (version 2; 9 km) | NEI 2016 beta |
| NEI14 | 12 km/63 layer | | none | NEI 2014 |
| Minus001 | | 20 and 27 of August 2016 (ACT-America) | none, surface SM initial conditions reduced uniformly by 0.01 $m^3m^{-3}$ across the domain | NEI 2016 beta |
| SEACf | 25 km/27 layer | 12-24 August 2013 (SEAC$^4$RS) | none | NEI 2014 |
| SEACa | | | ESA CCI passive (version 04.5; 0.25°) | NEI 2014 |

Acronyms: ACT: Atmospheric Carbon Transport; ESA CCI: European Space Agency Climate Change Initiative; NEI: National Emission
Inventory; SEAC$^4$RS: Studies of Emissions and Atmospheric Composition, Clouds and Climate Coupling by Regional Surveys; SM: Soil Moisture; SMAP: Soil Moisture Active Passive; WRF-Chem: Weather Research and Forecasting model with online Chemistry

**Table 2: The SMAP DA impacts on modeled surface meteorological and O₃ fields, as well as their agreement with observations.**

| Variable analyzed | Assim-Base case, domain mean ± standard deviation, for all overland grids | RMSE, Base case, domain mean ± standard deviation | ΔRMSE, Assim-Base case, domain mean ± standard deviation | % of the model grids with available observations in which the SMAP data assimilation improved the model performance |
|---|---|---|---|---|
| Daytime 2 m air temperature | $0.099 \pm 0.373$ K | $2.177 \pm 0.718$ K | $\sim 0 \pm 0.165$ K | 47.2% |
| Daytime 2 m relative humidity | $-0.573 \pm 3.225$ % | $12.633 \pm 4.188$ % | $-0.024 \pm 1.765$ % | 51.3% |
| Daytime 10 m wind speed | $0.001 \pm 0.129$ ms$^{-1}$ | $1.714 \pm 0.831$ ms$^{-1}$ | $-0.005 \pm 0.183$ ms$^{-1}$ | 52.5% |
| MDA8 O₃ | $0.141 \pm 0.494$ ppbv | $7.674 \pm 2.473$ ppbv (referring to AQS); $6.710 \pm 2.285$ ppbv (referring to CASTNET); | $0.057 \pm 0.372$ ppbv (referring to AQS); $0.007 \pm 0.343$ ppbv (referring to CASTNET); | 42.0% (referring to AQS); 51.4% (referring to CASTNET) |

Acronyms: AQS: Air Quality System; CASTNET: Clean Air Status and Trends Network; MDA8: daily maximum 8-h average; RMSE: root-mean-square error; SMAP: Soil Moisture Active Passive

**Table 3: The number of model grids with surface MDA8 O₃ exceedance false alarms (i.e., the modeled MDA8 O₃>70 ppbv but the observed MDA8 O₃<=70 ppbv) from three 12 km simulations which are defined in Table 1. Degradations and improvements from the base case are highlighted in italic and bold, respectively.**

| Days of August 2016 | Referring to AQS observations | | | Referring to CASTNET observations | | |
|---|---|---|---|---|---|---|
| | Base | Assim | NEI14 | Base | Assim | NEI14 |
| 16 | 0 | 0 | 0 | 0 | 0 | 0 |
| 17 | 0 | 0 | 0 | 0 | 0 | 0 |
| 18 | 1 | *3* | *4* | 0 | 0 | 0 |
| 19 | 9 | 9 | *10* | 0 | 0 | 0 |
| 20 | 4 | 4 | *13* | 0 | 0 | *1* |
| 21 | 0 | 0 | 0 | 0 | 0 | 0 |
| 22 | 0 | 0 | 0 | 0 | 0 | 0 |
| 23 | 1 | 1 | 1 | 0 | 0 | 0 |
| 24 | 1 | 1 | *2* | 0 | 0 | 0 |
| 25 | 1 | 1 | *2* | 0 | 0 | 0 |
| 26 | 6 | **5** | *9* | 1 | **0** | 1 |
| 27 | 0 | 0 | 0 | 0 | 0 | 0 |
| 28 | 6 | 6 | *14* | 0 | 0 | 0 |

Acronyms: AQS: Air Quality System; CASTNET: Clean Air Status and Trends Network; MDA8: daily maximum 8-h average

**Figures**

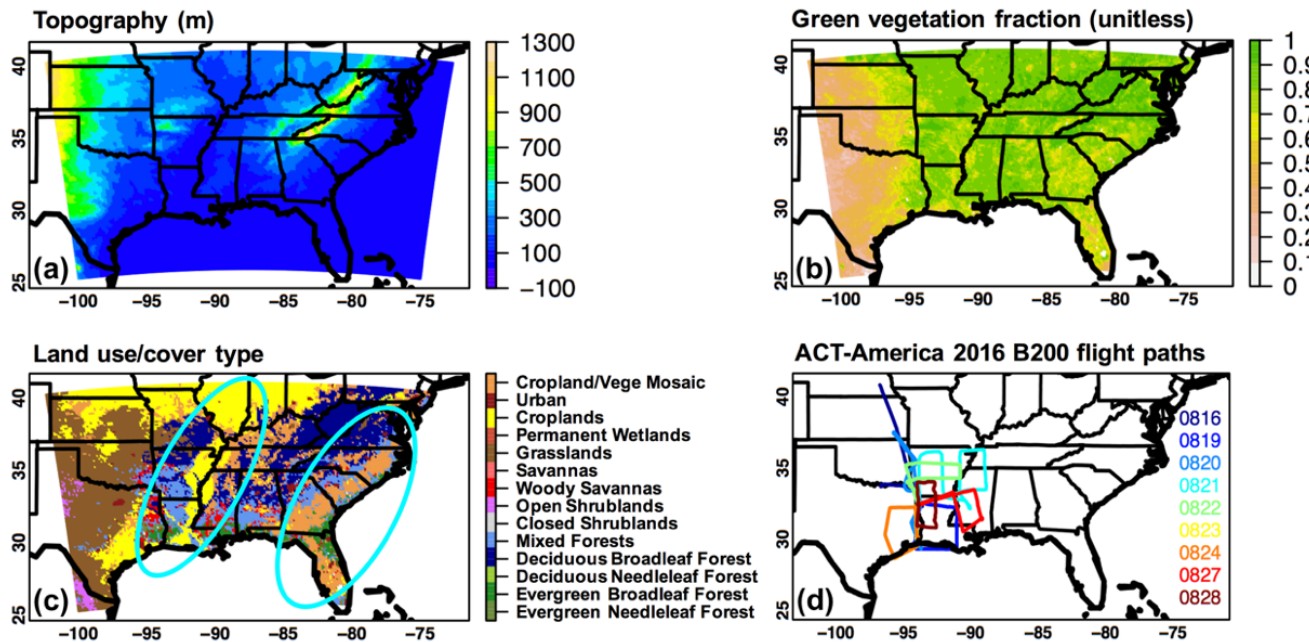

Figure 1: (a) Terrain heights; (b) August 2016 green vegetation fraction; and (c) grid-dominant land use/cover categories used in the 12 km LIS/WRF-Chem simulations. (d) B-200 flight paths in the southeastern US during the 2016 ACT-America campaign. Cyan-blue circles in (c) denote the approximate locations of areas with high irrigation water use based on literature. Similar model domains, consistent sources of geographical inputs and meteorological forcings were used in 12 km and 25 km LIS/WRF-Chem simulations.

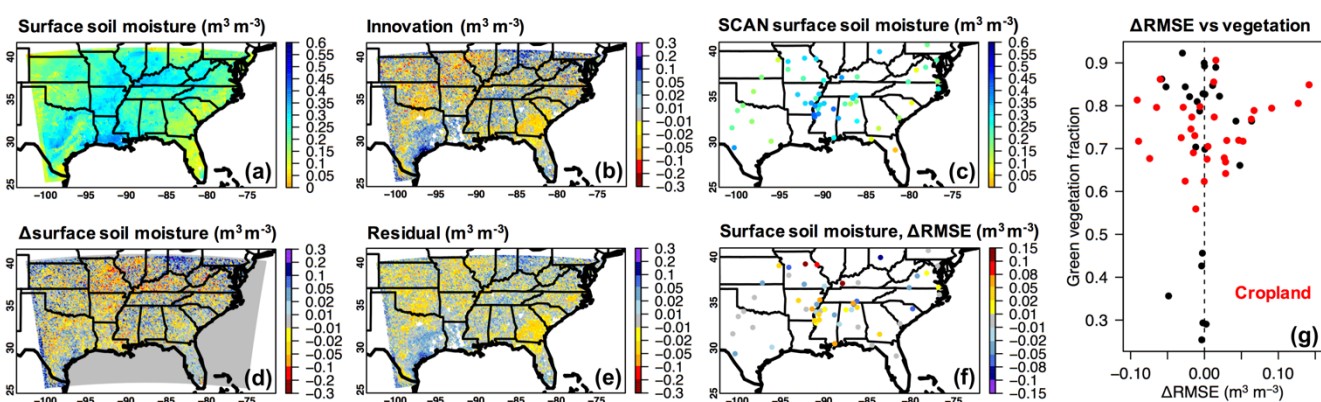

Figure 2: Period-mean (16-28 August 2016) (a) WRF-Chem base case surface-layer (i.e., 0-10 cm belowground) soil moisture at initial times and (d) its changes due to the SMAP DA. (b;e) indicate the SMAP DA impacts on the discrepancies between SMAP and modeled surface soil moisture. (c) presents soil moisture measurements at various SCAN sites at ~10 cm belowground at WRF-Chem initial times. The SMAP DA impacts on RMSEs of the modeled surface soil moisture, as well as their relationships with the model-based green vegetation fraction, are shown in (f-g). In (g), the SCAN sites located in cropland areas according to the model's land use/cover input are highlighted in red.

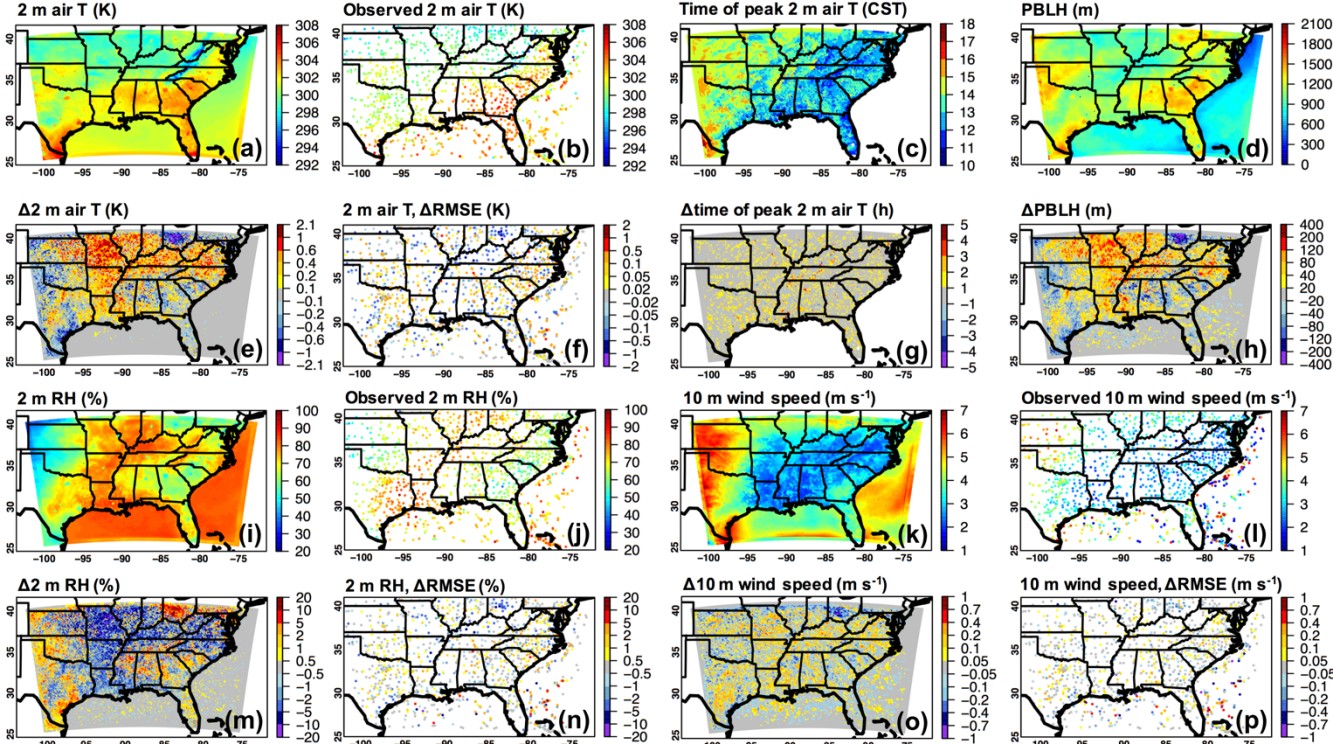

**Figure 3: Period-mean (16-28 August 2016) WRF-Chem base case daytime (a) 2 m air temperature (T); (d) PBLH; (i) 2 m relative humidity (RH); (k) 10 m wind speed, as well as (e;h;m;o) the impacts of SMAP DA on these model fields. Observed daytime surface T, RH and wind speed, as well as the impacts of the SMAP DA on RMSEs of these model fields are shown in (b;f), (j;n), and (l;p) respectively. Significance test results are included in Figure 4. The time of daily peak air T in US Central Standard Time (CST), as well as its response to the SMAP DA, is shown in (c;g).**

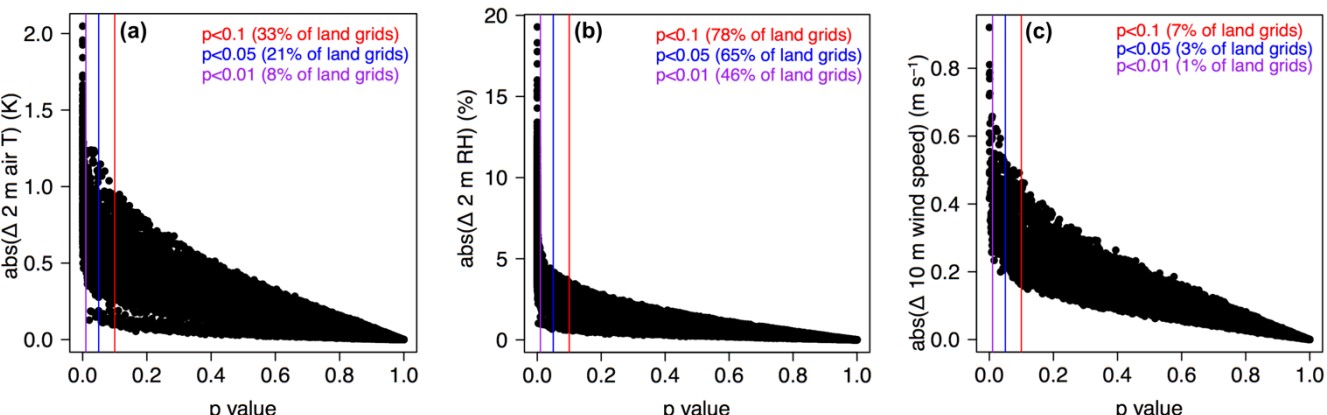

**Figure 4. The p values of the Student's t-tests comparing the daytime (a) 2 m air temperature (T); (b) 2 m relative humidity (RH); and (c) 10 m wind speed from the base and "assim" cases, plotted against the absolute changes in these model fields due to the SMAP DA. Results are only presented for the overland model grids.**

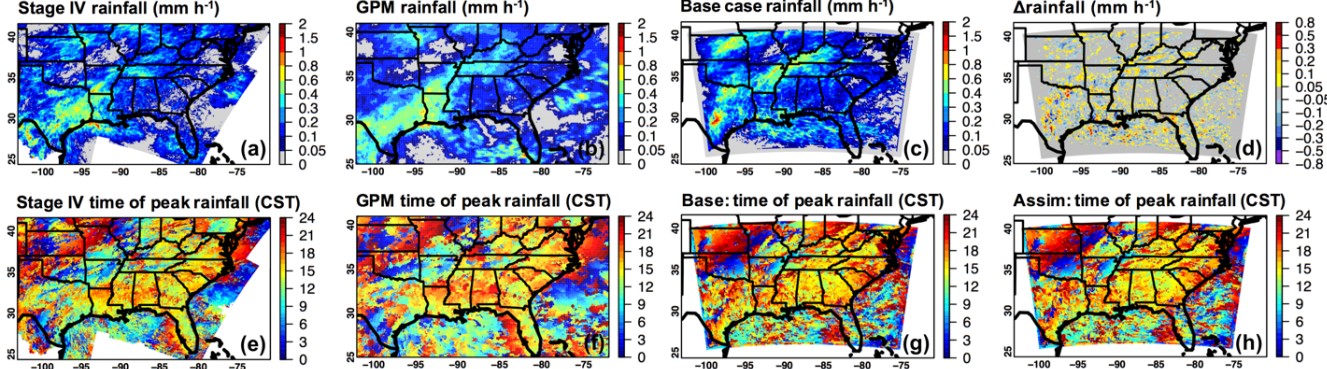

**Figure 5:** Period-mean (16-28 August 2016) (a-d) rainfall rate and (e-h) time of peak rainfall in US Central Standard Time (CST) from (a;e) the national Stage IV Quantitative Precipitation Estimates product; (b;f) the Global Precipitation Measurement; and (c;g) WRF-Chem base case. The impacts of the SMAP DA on WRF-Chem results are indicated in (d;g-h).

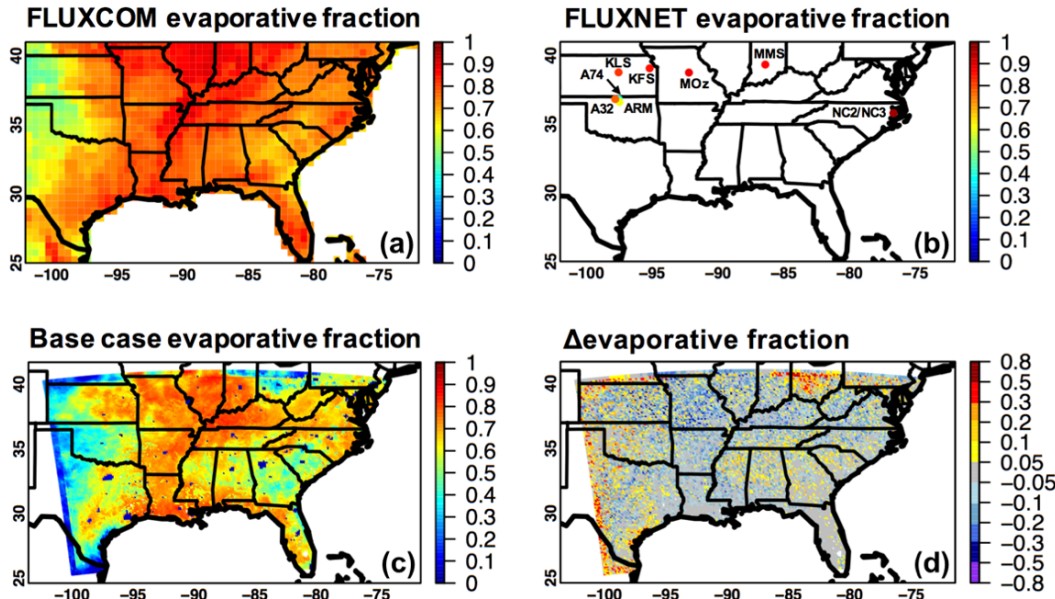

**Figure 6:** Period-mean (16-28 August 2016) daily evaporative fraction, defined as: daily latent heat/(daily latent heat+daily sensible heat), from (a) a FLUXCOM product; (b) selected FLUXNET sites; and (c) WRF-Chem base case. (d) shows the impact of the SMAP DA on WRF-Chem EF. Additional evaluation results for latent and sensible heat fluxes at the focused FLUXNET sites are presented in Figure S3.

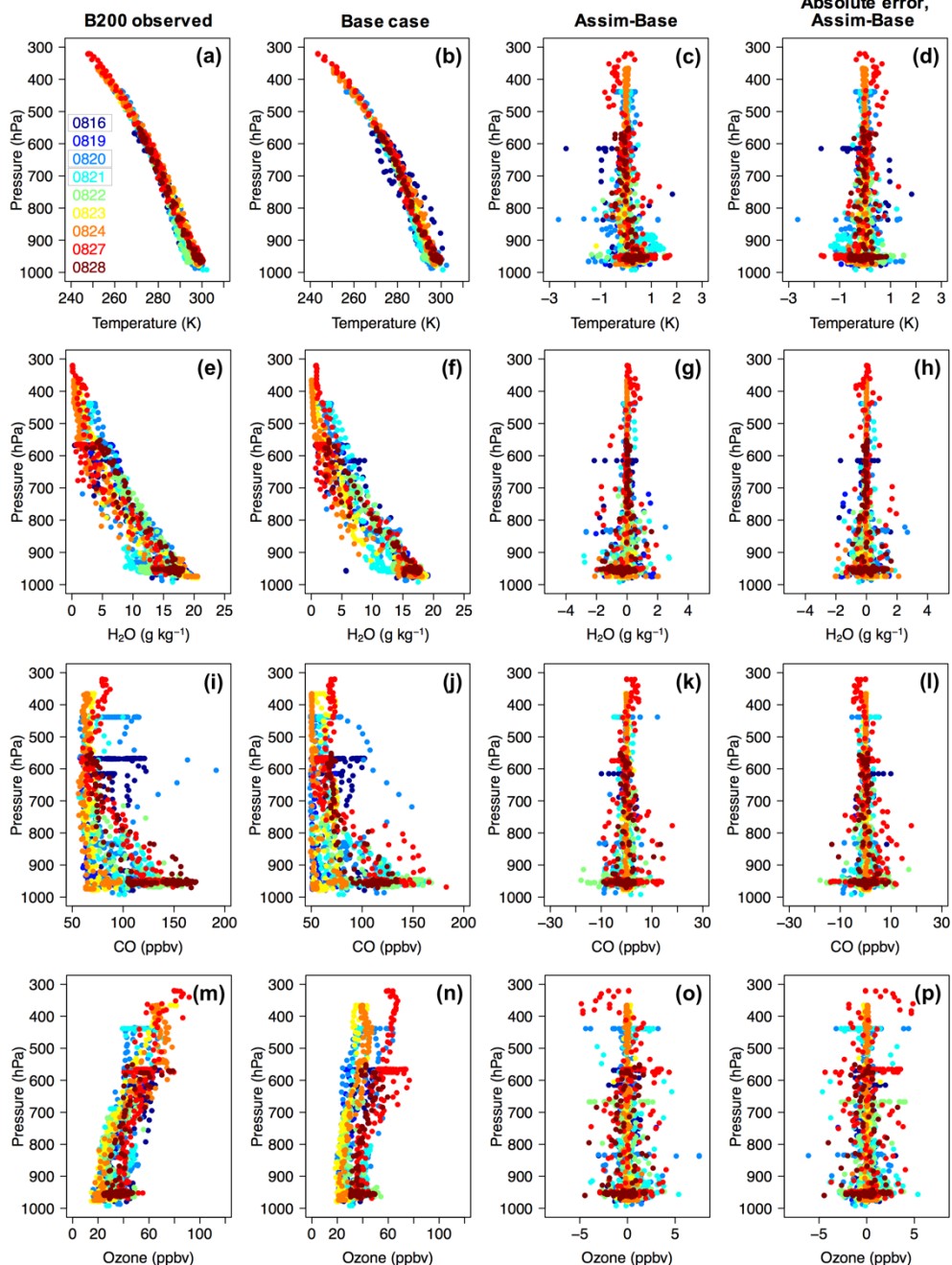

**Figure 7: Vertical profiles of (a) air temperature; (e) water vapor mixing ratio (H₂O); (i) carbon monoxide (CO); and (m) O₃ observed on the B-200 aircraft during the ACT-America 2016 campaign, based on a 1-minute averaged dataset. Their WRF-Chem counterparts from the base case and the impacts of the SMAP DA are shown in (b;f;j;n) and (c;g;k;o), respectively. The SMAP DA impacts on model performance along these flights, based on the absolute error metric (i.e., |modeled-observed|), are indicated in (d;h;l;p). The different colors distinguish samples taken on various flight days, and the B-200 paths on these flight days are shown in Figure 1d. Flights on 16, 20, 21 of August 2016 were conducted under stormy weather conditions as highlighted in (a), whereas the B-200 flew under fair weather conditions during other flights.**

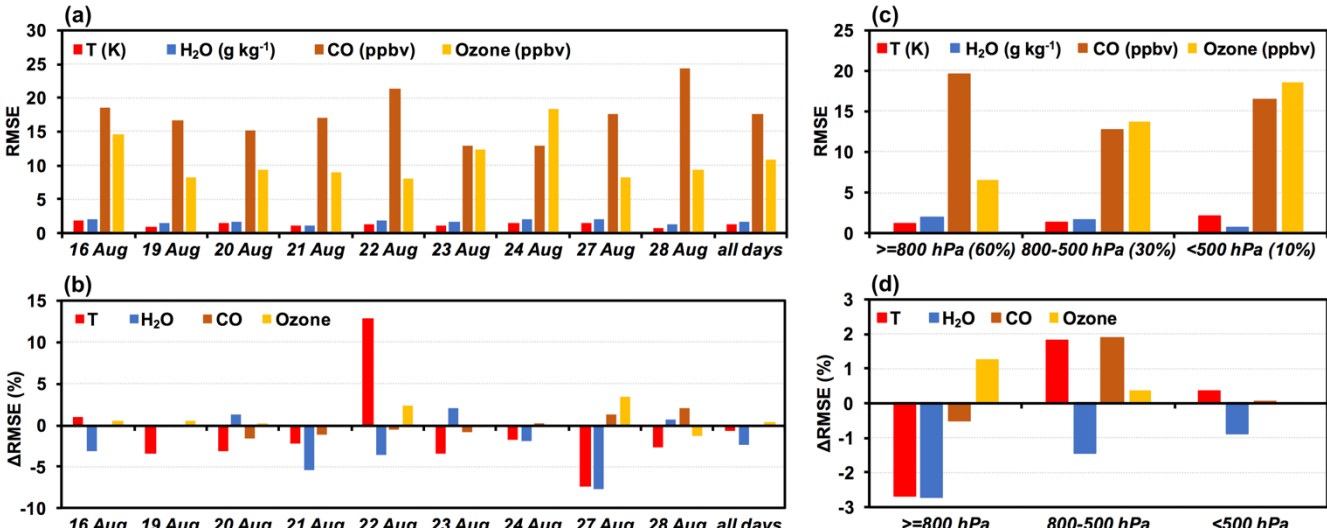

**Figure 8: Evaluation of WRF-Chem results with the B-200 aircraft observations during the ACT-America 2016 campaign: (a;c) the RMSEs of air temperature (T), water vapor mixing ratio (H$_2$O), carbon monoxide (CO) and O$_3$ of the model base case; and (b;d) the impacts of the SMAP DA on RMSEs of these variables. (a-b) and (c-d) summarize the model performance by flight day and flight altitude range, respectively. The B-200 flight paths by day are shown in Figure 1d. ~60%, ~30%, and ~10% of the related aircraft observations were taken at >= 800 hPa, 800-500 hPa, and <500 hPa, respectively.**

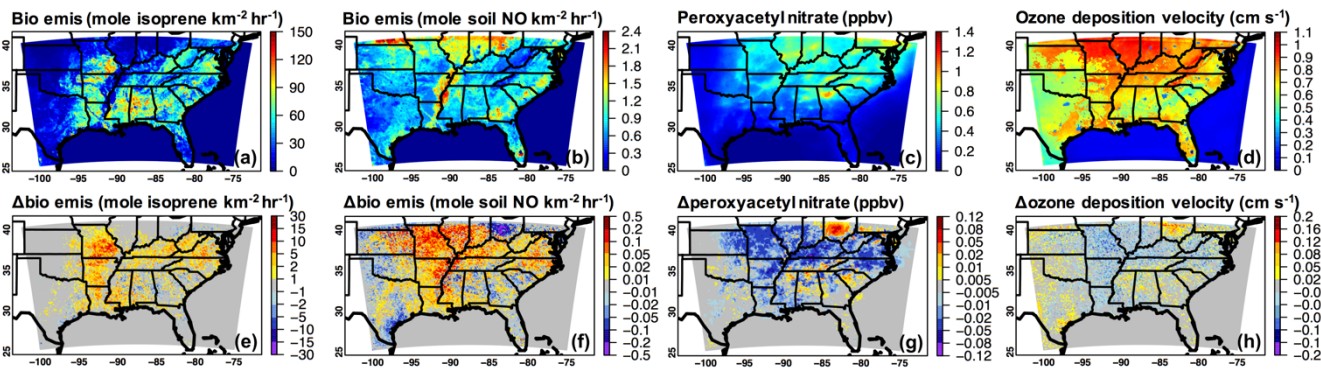

**Figure 9: Period-mean (16-28 August 2016) WRF-Chem base case daytime biogenic emissions of (a) isoprene and (b) soil nitric oxide (NO); (c) surface peroxyacetyl nitrate concentration; and (d) O$_3$ deposition velocity, as well as (e-h) the impacts of SMAP DA on these model fields. Additional results of these variables are shown in Figure S5.**

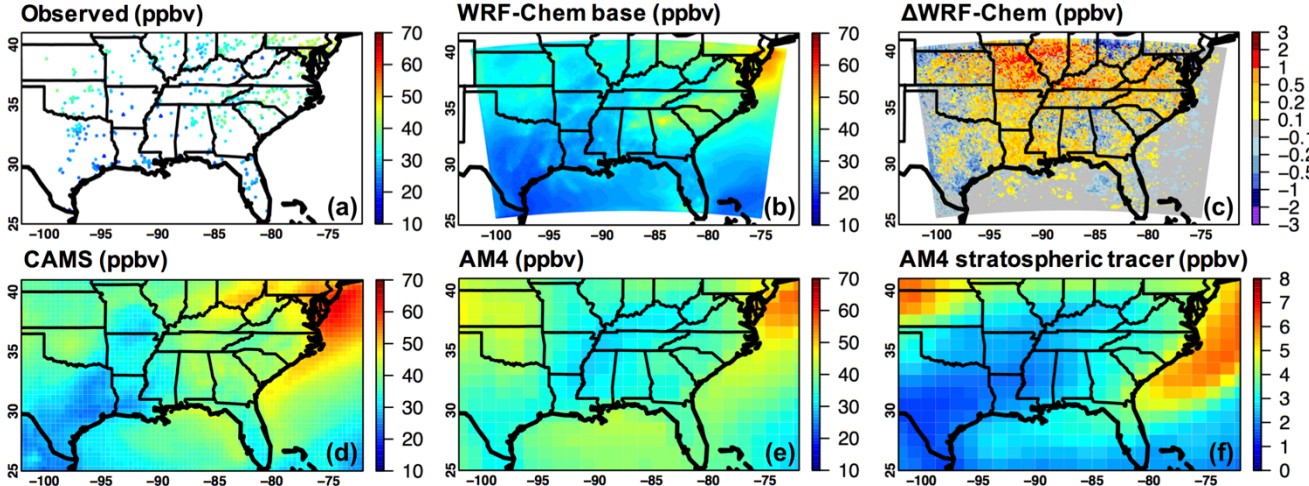

**Figure 10: Period-mean (16-28 August 2016) daytime surface O₃ from (a) the EPA AQS (filled circles) and CASTNET (triangles) sites; (b) WRF-Chem base case; (d) CAMS; and (e) GFDL AM4. (c) shows the impact of the SMAP DA on WRF-Chem modeled daytime surface O₃. (f) indicates stratospheric influences on daytime surface O₃ based on the AM4 stratospheric O₃ tracer output.**

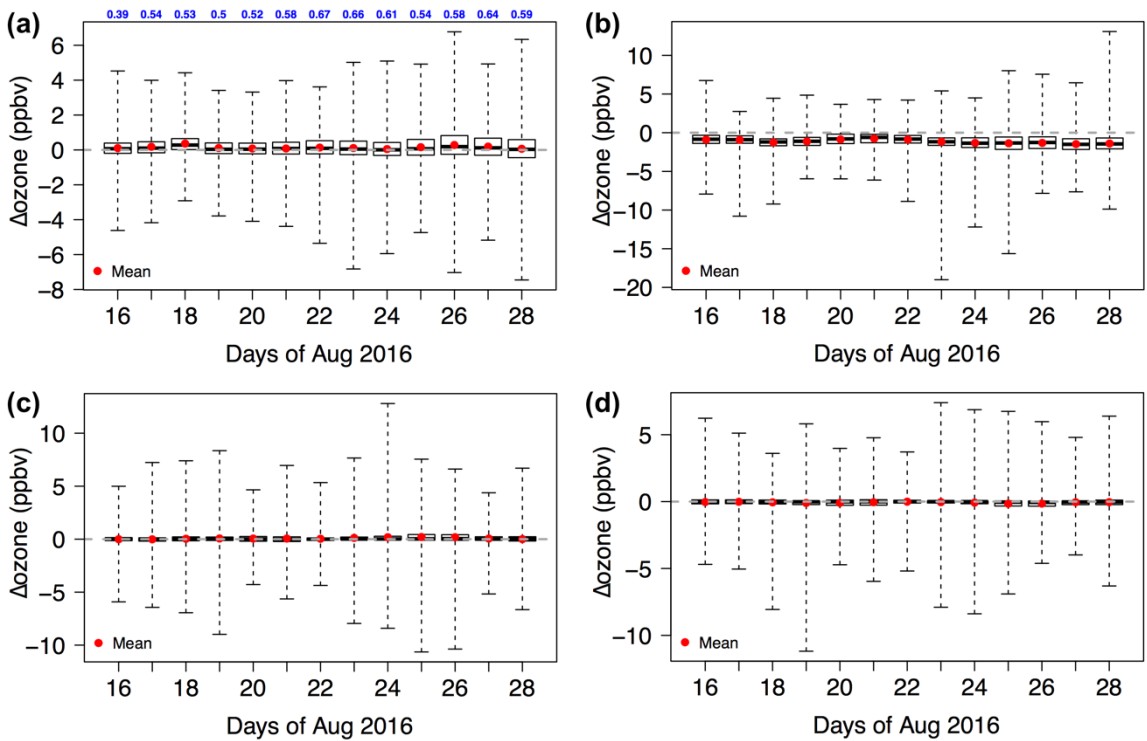

**Figure 11: Box-and-whisker plots of WRF-Chem daytime O₃ responses to (a;c) the SMAP DA; and (b;d) updating anthropogenic emissions from NEI 2014 to NEI 2016 beta. (a-b) and (c-d) show O₃ changes at the surface (only for terrestrial model grids, 68% of all model grids) and at ~400 hPa (in all model grids), respectively. Blue text in (a) are spatial correlation coefficients *r* between WRF-Chem daily daytime 2 m air temperature changes and O₃ changes due to the SMAP DA. Note the different Y-axis ranges.**

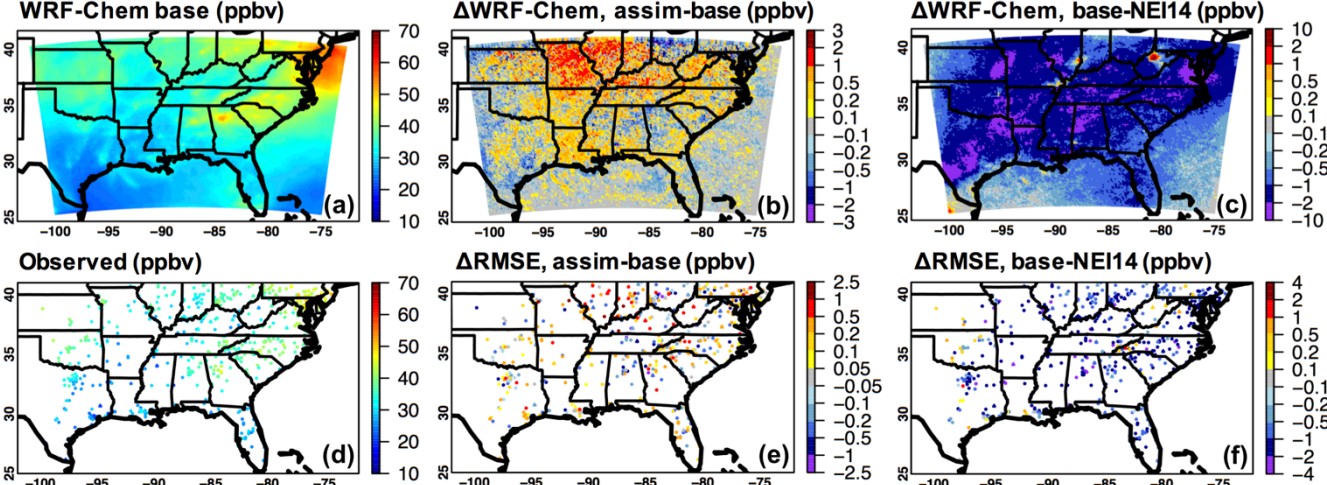

**Figure 12:** Period-mean (16-28 August 2016) daily maximum 8-h average (MDA8) surface $O_3$ from (a) WRF-Chem base case and (d) the EPA AQS (filled circles) and CASTNET (triangles) sites. The impact of the SMAP DA on WRF-Chem MDA8 $O_3$ and the associated RMSE changes are shown in (b) and (e), respectively. The benefit of using NEI 2016 beta instead of NEI 2014 is indicated in (c;f).

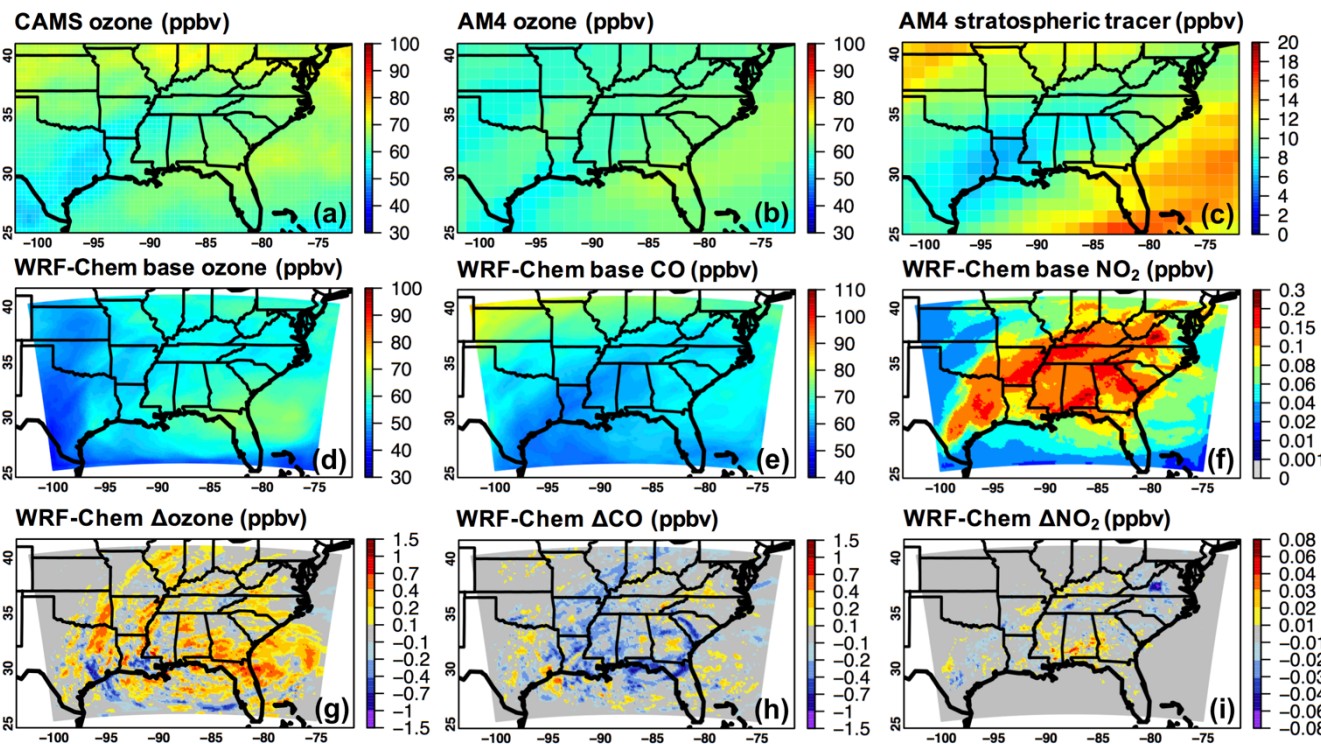

**Figure 13:** Period-mean (16-28 August 2016) daytime $O_3$ in the upper troposphere (i.e., the model levels close to 400 hPa) from (a) CAMS; (b) GFDL AM4; and (d) WRF-Chem base case. (g) shows the impact of the SMAP DA on WRF-Chem modeled daytime $O_3$ in the upper troposphere, and (c) indicates the stratospheric influences on $O_3$ at these altitudes based on the AM4 stratospheric $O_3$ tracer output. Period-mean daytime CO and $NO_2$ from WRF-Chem base case as well as their responses to the SMAP DA are shown in (e;h) and (f;i), respectively.