# Peer review of "Satellite soil moisture data assimilation impacts on modeling weather variables and ozone in the southeastern US - part I: an overview"

_Atmospheric Chemistry and Physics, 2020_

## Referee Comment (RC1) · Anonymous Referee #2 · 18 Sep 2020

This paper describes the impacts on the representation of meteorological variables and ozone in the southeastern US in WRF-Chem of assimilating soil moisture into the Noah land surface model. It demonstrates that soil moisture has an influence on these variables and provides a useful indication of the magnitude of the effects. The paper addresses an interesting topic, shows elements of novelty, is mostly of satisfactory quality and is within the defined scope of ACP.

My principal criticism is that while it is an interesting and competent description of a sensitivity experiment on soil moisture, with justification and explanation of results, it does not in its present form provide the analysis and deeper insight needed to substantially improve current understanding. This is largely because the focus is on the effects of assimilation rather than on the wider effects of soil moisture on the model atmosphere. This provides little new process understanding, may not be applicable to other models, and depends heavily on the performance of the underlying Noah land surface model, which is not explored in any detail here. While it is clear that this is an exploratory study, frequent statements in the results and discussion such as "future efforts should be devoted to..." and "... need further evaluation" point to topics that should have been explored more thoroughly here. This is particularly the case where key processes or feedbacks are acknowledged to be missing (e.g., soil moisture controls on VOC emissions from MEGAN, or on deposition processes and vegetation uptake).

The quality of the data assimilation needs to be assessed more thoroughly before the atmospheric impacts can be explored. If data assimilation of soil moisture has a large effect it suggests that there are either substantial biases in the Noah land surface model or major uncertainties in the retrieved values. This uncertainty needs to be summarized to aid the reader in interpreting the results.

Much of the paper is descriptive rather than analytic, and this needs to be addressed before the paper is suitable for publication. The methods section in particular is too long. The results section describes comparisons, supported by a large number of figures, but the explanations are largely speculative and provide little new insight into the governing processes. The comparison with aircraft observations is somewhat cursory, and given that the improvements may not be significant (although this is not assessed rigorously) then it is not clear what value the comparisons bring.

The sensitivity study on anthropogenic emissions (Section 3.5) does not fit well with the main focus of the study on soil moisture, and it is not clear why this was included. I would recommend removing this section and the associated comments in the conclusions (lines 538-542) which are of little relevance to data assimilation of soil moisture.

The paper concludes by investigating the impacts on the upper troposphere and potential effects downwind. While it is valuable to explore the wider implications of soil moisture assimilation, the effects on ozone are very small (less than 1 ppb) and are much less than the biases associated with poor representation of stratospheric contributions due to lack of upper boundary conditions. The value and significance of this comparison is therefore unclear. This should be established before the potential consequences for ozone over distant regions such as Europe is considered.

In summary, the paper needs some reformulation to bring out key messages. The weaknesses identified here could be addressed in a number of ways. A simple sensitivity study altering soil moisture uniformly across the domain could be very useful to confirm the impact on different processes (e.g., lightning, convection) and would allow a more authoritative interpretation of the complexity of varying biases associated with assimilation. Tightening the methods and results sections by replacing description with explanation or analysis would be helpful. Further specific comments and suggestions are included below.

The English language is acceptable but is awkward in places, and the text would certainly benefit from some polishing.

Specific Comments

Title: the paper addresses the impacts on meteorological variables, not on "weather" in a conventional sense, and the title should be adjusted to reflect this.

Abstract, line 17: "dense vegetation, complex terrain, unmodeled water use" These issues are included in the abstract, section 3.2 and conclusions but are results from previous work, not the outcome of analysis in the present study.

Abstract, lines 23-27: These two sentences should be rephrased. The focus needs to be on the importance of the processes rather than the importance of quantifying them, and accurate assessment of the SMDA impacts on model performance is less important than understanding the importance of correctly-represented SM.

Line 59: clearer phrasing is needed: trapping in the upper troposphere rather than anticyclones established there?

Line 65: Soil moisture has other influences on the atmosphere (e.g. indirectly through vegetation) so perhaps add "principally" or "most greatly" here.

Line 81: The term "semicoupled" is not meaningful, as it remains unclear which components are coupled and which are not. Is this a form of one-way coupling or a coupling of only some variables? A clear but concise description is needed to explain this to the reader.

Line 125: What is the justification for the bias correction described here, and how much impact does it have?

Line 167: If soil moisture influences are not well represented in Megan, will the responses to its assimilation be meaningful or useful? The effects are only indirect through other meteorological variables. Similarly, what are the consequences of the lack of VPD treatment in the deposition scheme? This is only briefly mentioned in the text at l.400.

Line 169: "curves" would be clearer as "vertical profiles"

Para 230: Are these observations published? If so, please provide citations.

Line 331: A table of model performance with and without DA is needed here to provide a stronger quantitative underpinning of this discussion.

Line 343-345: There is no clear signal from the assimilation of a bias associated with irrigation in the regions indicated; why is this? Is this difference swamped by other uncertainties, or is the effect washed out by the bias correction applied before assimilation?

Line 356-358: this explanation for model problems with evaporative fraction is vague and unconvincing!

Line 367: The impacts of the data assimilation on temperature and humidity are very small. Are these changes significant?

Fig 6 shows the observations, the model simulation and the impacts of assimilation. However, it does not show whether the base simulation matches the observations or whether the assimilation improves the model bias, and these are the two factors that the reader is most interested in! Some of this information is provided in Figure 7 on a temporal not vertical basis, but please reconsider which panels to show in Fig 6.

Line 405: It would be worth pointing out that these RMSE changes are positive and that model performance is less good with assimilation.

Line 426: The points made in this paragraph highlight compensating model errors for ozone, but the lack of any stratospheric influence in the WRF-Chem runs remains an issue to be addressed.

Line 451: lightning is mentioned in the abstract, conclusions and a number of places through the paper, but the effects are not quantified anywhere. Does the soil moisture assimilation have any significant effect on lightning NO emissions? If so, please quantify it.

Line 495: Evaluation against SEAC4RS observations is not thorough here. Assimilation "led to better model agreements" but no numbers are provided in support of this. Some indication of the biases or RMSE values are needed in the text or a table, or alternatively a scatter plot of simulations against observations should be added to Figure S5. While this attempt to put the results of the study in context is valuable, the comparison is not convincing, and the explanations are highly speculative.

Line 510: Improvements in T2/RH/WS in 50% of locations is not a convincing demonstration of the value of assimilation. The improvements in MDA8 against AQS and CASTNET (42%, 51%) are of very similar (negligible?) magnitude, but these details are omitted from the concluding discussion.

Fig S1: The panels in this figure are too small, please make them larger so that they are legible (as in Fig 1).

Typos and Minor Issues

The language needs substantial polishing, e.g., line 103 "of the used modeling system" better as "of the modeling system used". (and Line 341)

Line 111: acronym SRTM30 is not defined. Line 139: is -> are Line 340: better phrased more clearly without use of "unmodeled"

---

## Referee Comment (RC2) · Anonymous Referee #1 · 19 Oct 2020

This study addresses the impact of a more accurate treatment of soil moisture content on WRF-Chem simulations of some aspects of weather and atmospheric composition. The soil moisture of the NOAH land surface model is adjusted using a data assimilation technique to retrieved soil moisture content from the NASA SMAP radiometric measurements. The more accurate soil moisture then modifies moisture, heat and trace gas emissions from their 'base' values, and two WRF-Chem studies are compared for the period of August 2016, one being the base and the other including SM DA. Additional simulations are performed for 2013. Comparisons are between ground- and aircraft-based observations and modelled quantities.

This is a competent modelling study, and the authors have attempted to apply best practice in bringing SM DA and the application of WRF Chem to the study of the continental US. As such it complements, but doesn't much extend, an earlier study by the lead author in 2018. It therefore somewhat lacks novelty.

The SM DA is shown to improve model performance as compared to aircraft observations of air temperature and specific humidity, although there is no reported improvement against ground-based observations of temperature, humidity or wind speed.

For reactive gas phase composition, very small changes in O3 are calculated, with little effect of SM DA on modelled ozone aloft. Some degradation in model skill results, which the authors phrase as being less 'desirable'. I think this means they expect SM DA to improve model skill, but as it stands there are no reasons in the manuscript given.

The study concludes with the effect of including an updated emissions database on modelled ozone.

I would identify this as an interesting region/time period for study being a geographical region with a heterogeneous LULC environment results where there are multiple sampling of edge cases (regions of drought, regions close to field capacity) in the the vegetation modelling framework. .

My main issue with the MS is that it's something of a pot-boiler, and the problem under consideration is not clearly stated. The study is undermined by the majority of the discussion being rather qualitative, despite much quantitative information being in the paper's figures, and the discussion is often focussed on what was not included, rather highlighting the impact of SM DA on model performance.

Some important questions are raised, but no clear direction of travel for this work emerges and the no clear conclusions are drawn as to how and to what extent SM modifies the picture until we reach the concluding remarks. This diminishes its impact and I suggest that the focus of any revised submission should be on the process-level

impacts of SM DA on e.g. emissions or deposition processes which result from the improved treatment of SM.

I say this because, at present, the authors bring up aspects of the modelling framework which are unsatisfactory or where the study itself could have been improved and some readers might be left wondering exactly what remains of the SM impact that has been included at the process level. Slightly frustratingly, there are long parts of the discussion on things that can't be addressed (340-345 irrigation. 346-347 rainfall product QC, 358-360 other models that might be used). L210 raises the question of how SM affects convection, but no discussion of the impact is given on e.g. vertical transport.

As a second example, on L167 the critically important aspect of the response of MEGAN to SM is raised, but, after saying that SM effects are not well understood, the discussion moves on, although figure 3 shows the modifications. The authors need to extend this section extensively to quantify the effect of SM on emissions in MEGAN, and to show how MEGAN responds (especially as these NO and isoprene emissions changes are important to the discussion of the ozone response to SM DA).

As a third, L187 raises SM-dependent vegetation properties which might be important to ozone deposition, but it is again not clear what the impact of these effects might be on e.g. ozone. These issues are raised again L379 and again in L396-403 with similar lack of clarity as to their impact. If these important effects can't be at present included, it seems to me that having raised these points repeatedly, the authors should at least estimate what the size of these effects might be on deposition velocity and hence ozone flux to the surface? The manuscript would be improved drastically if these important processes were discussed quantitatively - above, it would be good to know how big are the emissions changes. Here, how large would deposition velocities need to change to produce an effect on ozone?

As a fourth example, the study makes a point about how the signal from the use of different emissions datasets in terms of ozone response is large. This is a worthy point,

but does not contribute to the question at hand, and the use of a second emissions dataset doesn't really improve the understanding of the problem. Similarly, the role of strat-trop transport is undoubtedly important, but again moves the discussion away from the SM DA. The reluctance to exclude anything, and to state which factors are dominant, makes the focus of the study very difficult to discern, and really detracts from the potential impact which is to understand how the WRF-Chem modelling framework is improved by SM DA in this mixed LULC environment.

The ozone response appears to be driven by temperature via the coupling of MEGAN to WRF meteorology. Here the manuscript is somewhat successful but this section also gives the clearest indication of how it could be improved. The authors could expand on the description of the results to drill down into the processes at work and how they interact. For instance, Figure 9 shows a very small change in ozone, which receives little comment, the authors preferring instead to concentrate on the maximum value and the correlation. The revised MS could look at regions of positive or negative ozone change, and say whether the small change in O3 is to be expected, or not, and give reasons for this, for instance by unpicking the contribution from emissions, deposition and temperature changes in Figure 3, and to present the results in more detail than is done in L389-392. This approach should be followed for the other aspects of the impact of SM DA on O3 and other atmospheric constituents.

In conclusion, I feel that the impact of the study would be improved if the focus could be narrowed, the depth of discussion improved and the connection of SM to the other inputs to WRF-Chem better quantified.

Specific comments:

L43: missing symbol between 70 ppbv

L55: mid-latitude cyclones are

L56: 'They are'

[Figure]

L138-139: 'the major chemical species in the FT are'

L148: what do the authors mean by 'runs'?

L180: rephrase 'its major component surface resistance'

L307: shown to be consistent

L330: unusual use of dominantly

L533: sentence describing the impact is not clear

---

## Author Comment (AC1) · 4 Jan 2021

**Author response to reviews**

The authors appreciate both referees' careful reviews and constructive comments, which helped us improve this manuscript significantly. Please see below our point-to-point response (in blue) to both referees' general and specific comments (in black). Quoted text from the revised manuscript is *in italic*.

**Response to RC1**

This paper describes the impacts on the representation of meteorological variables and ozone in the southeastern US in WRF-Chem of assimilating soil moisture into the Noah land surface model. It demonstrates that soil moisture has an influence on these variables and provides a useful indication of the magnitude of the effects. The paper addresses an interesting topic, shows elements of novelty, is mostly of satisfactory quality and is within the defined scope of ACP. My principal criticism is that while it is an interesting and competent description of a sensitivity experiment on soil moisture, with justification and explanation of results, it does not in its present form provide the analysis and deeper insight needed to substantially improve current understanding. This is largely because the focus is on the effects of assimilation rather than on the wider effects of soil moisture on the model atmosphere. This provides little new process understanding, may not be applicable to other models, and depends heavily on the performance of the underlying Noah land surface model, which is not explored in any detail here. While it is clear that this is an exploratory study, frequent statements in the results and discussion such as "future efforts should be devoted to..." and "... need further evaluation" point to topics that should have been explored more thoroughly here. This is particularly the case where key processes or feedbacks are acknowledged to be missing (e.g., soil moisture controls on VOC emissions from MEGAN, or on deposition processes and vegetation uptake).

Thank you for the overall positive feedback and the suggested revisions. The Noah land surface model used in this study has long been, and is still, widely used in land, weather and air quality modeling communities. Therefore, we believe that case studies using Noah are informative to various audiences. Further investigations have been conducted. The revision more clearly explains how specific limitations of Noah affect the results and conclusions, and how the results would look like if the modeling experiments were conducted with certain processes treated differently in the model. Some of the Noah-related limitations can be addressed by recalibrating selected key model parameters using laboratory/field data or/and applying a different model (e.g., Noah-MP with dynamic vegetation). The Noah-MP based results already exist, which support our discussions on the Noah-based results in this revised manuscript, and they will be presented separately. This referee explicitly suggested including a sensitivity simulation with a constant SM perturbation. This suggestion has been taken and please refer to our response to that specific comment for details.

The quality of the data assimilation needs to be assessed more thoroughly before the atmospheric impacts can be explored. If data assimilation of soil moisture has a large effect it suggests that there are either substantial biases in the Noah land surface model or major uncertainties in the retrieved values. This uncertainty needs to be summarized to aid the reader in interpreting the results.

DA diagnostics during the case study period of ACT-America, such as innovations and residuals, are now included in the SI and also shown below. Other diagnostics such as statistical distributions of normalized O-minus-F and evaluation with ground-based SM observations would not be as

helpful due to the short study periods and large mismatches between the model's and surface sites' spatial scales. Evaluating the modeled weather and surface fluxes also provided assessments of the effectiveness of the SM DA, and the model evaluation for these variables has been significantly extended accounting for both referees' comments.

[Figure]

Much of the paper is descriptive rather than analytic, and this needs to be addressed before the paper is suitable for publication. The methods section in particular is too long. The results section describes comparisons, supported by a large number of figures, but the explanations are largely speculative and provide little new insight into the governing processes. The comparison with aircraft observations is somewhat cursory, and given that the improvements may not be significant (although this is not assessed rigorously) then it is not clear what value the comparisons bring.

This comment has been addressed via: 1) condensing Sections 2 and 3.5 (which has been merged into Section 3.4) as well as adding Section S1 and moving Figure 12 to Figure S9; 2) adding Table 2 as well as modifying Figures 6, 7, and S10 (previous S5) to more clearly and quantitively present the changes in model fields and the associated model performance changes across three dimensions; 3) adding information based on supporting variables (e.g., vertical wind W, lightning $NO_x$ tracer), significance test results, and diagnostic metrics to the SI; 4) adding new analysis based on a constant SM perturbation simulation as suggested by this referee, which helped confirm the SM influences on atmospheric weather and chemical fields at various locations on different flight days during the airborne campaign; and 5) extending the explanations on the model limitations related to biogenic emissions, dry deposition and surface fluxes, avoiding speculative language. Please also refer to our responses to the referees' specific comments.

The sensitivity study on anthropogenic emissions (Section 3.5) does not fit well with the main focus of the study on soil moisture, and it is not clear why this was included. I would recommend removing this section and the associated comments in the conclusions (lines 538-542) which are of little relevance to data assimilation of soil moisture.

The previous Section 3.5 (which has been merged into Section 3.4) and conclusions at L538-542 have been substantially modified. Please also see the response to your next comment regarding comparing the changes in UTLS $O_3$ due to the SM DA with those due to the NEI anthropogenic emission update.

The paper concludes by investigating the impacts on the upper troposphere and potential effects downwind. While it is valuable to explore the wider implications of soil moisture assimilation, the effects on ozone are very small (less than 1 ppb) and are much less than the biases associated with poor representation of stratospheric contributions due to lack of upper boundary conditions. The value and significance of this comparison is therefore unclear. This should be established before the potential consequences for ozone over distant regions such as Europe is considered.

Please note that SM DA impacts on UTLS $O_3$ are even more strongly variable in space and time than the impacts on surface conditions, and the magnitudes of ~1 ppbv or less are for averaged

results. As Figures 6, 7, and 9 show, during individual events, such impacts can reach as large as >10 ppbv sometimes. The response to your next comment also demonstrates that the SM DA on upper tropospheric composition can be very intense only in small fractions of the entire model domain. Furthermore, the changes in UTLS $O_3$ due to the SM DA are compared with those due to the NEI anthropogenic emission update (Figure 9), and the latter approach is often used to evaluate the benefits of US emission reductions to air quality in the downwind areas at different timescales regardless the model representations of the stratospheric influences. This comment is addressed via: 1) extending event-scale analysis; 2) reorganizing Sections 3.4-3.5 and moving Figure 12 to Figure S9; and 3) avoiding explicit comments on European $O_3$ pollution because this is a regional-scale modeling study and our domain does not cover Europe.

In summary, the paper needs some reformulation to bring out key messages. The weaknesses identified here could be addressed in a number of ways. A simple sensitivity study altering soil moisture uniformly across the domain could be very useful to confirm the impact on different processes (e.g., lightning, convection) and would allow a more authoritative interpretation of the complexity of varying biases associated with assimilation. Tightening the methods and results sections by replacing description with explanation or analysis would be helpful. Further specific comments and suggestions are included below.

The paper has been reformulated accounting for both referees' comments. A sensitivity simulation with initial conditions of surface SM reduced by 0.01 $m^3$ $m^{-3}$ across the domain was conducted for two of the ACT-America flight days when weather conditions differed significantly. Key results from this new simulation are now included in the SI and some of them are also shown on the right. These added results indicate that convection associated with lighting,

[Figure]

sometimes with fronts involved, lifted CO to as high as <500 hPa above some locations, and that a change in SM had influences on these processes. We agree with this referee that including such sensitivity analysis is "very useful to confirm the impact on different processes (e.g., lightning, convection)..". We have made it clear to the readers that, in reality, it is not just the magnitude of SM, but also its spatial heterogeneity, that strongly affects the SM-convection-lightning feedbacks. Constraining the models' SM fields with observations, despite the various limitations mentioned, adjusts the magnitude and spatial heterogeneity of SM. The readers may refer to current Figure 6i-l for SM DA impacts on CO vertical distributions during transport events.

The English language is acceptable but is awkward in places, and the text would certainly benefit from some polishing.

The text has been extensively edited. Awkward language has been replaced or removed.

Specific Comments

Title: the paper addresses the impacts on meteorological variables, not on "weather" in a conventional sense, and the title should be adjusted to reflect this.

We added "*variables*" after "*weather*". Overall, "soil moisture interactions with weather" has more often been used than "soil moisture interactions with meteorological variables" (e.g., https://smap.jpl.nasa.gov/science/applications), so "*weather*" is kept in the revision.

Abstract, line 17: "dense vegetation, complex terrain, unmodeled water use" These issues are included in the abstract, section 3.2 and conclusions but are results from previous work, not the outcome of analysis in the present study.

Findings from previous work are now stated in the introduction, which are also used to explain the results shown in this paper. Sentences like these in the abstract and conclusions have been revised to make it clear what other aspects this study focuses on discussing, e.g., the missing processes such as irrigation.

Abstract, lines 23-27: These two sentences should be rephrased. The focus needs to be on the importance of the processes rather than the importance of quantifying them, and accurate assessment of the SMDA impacts on model performance is less important than understanding the importance of correctly-represented SM.

These sentences have been rephased, reflecting the added/modified analysis in the revision. The abstract covers effectiveness of DA, its impact on various processes, and model performance.

Line 59: clearer phrasing is needed: trapping in the upper troposphere rather than anticyclones established there?

Changed to: "*upper tropospheric anticyclones…*"

Line 65: Soil moisture has other influences on the atmosphere (e.g. indirectly through vegetation) so perhaps add "principally" or "most greatly" here.

We modified this sentence to clarify that "evapotranspiration" includes plant transpiration. In the later sessions, we explain that the SM DA in this study did not update vegetation (GVF, LAI) and suggest that applications using land models with dynamic vegetation would be preferred in future studies.

Line 81: The term "semicoupled" is not meaningful, as it remains unclear which components are coupled and which are not. Is this a form of one-way coupling or a coupling of only some variables? A clear but concise description is needed to explain this to the reader.

In at least two places of the paper semicoupled is introduced, specifically: 1) at this line: "*The term "semicoupled" here is similar to "weakly-coupled", as opposed to "fully-" or "strongly-" coupled, which indicates that the SM DA within LIS influences WRF-Chem's land initial conditions*"; and 2) in Section 2.1, "*Each day's WRF-Chem meteorological outputs served as the forcings of the no-DA and DA LIS simulations, which produced land initial conditions for next day's WRF-Chem simulations*". This means SM DA is conducted during the land cycle with WRF-

Chem forcings. The land initial conditions of WRF-Chem during the subsequent atmospheric cycle (in which atmospheric observations may or may not be assimilated, and not for this study), are influenced by the SM DA effects during the land cycle. The variables related to land initial conditions are specific to the land surface model used. For Noah, the most important prognostic variables are soil moisture and soil temperature.

Line 125: What is the justification for the bias correction described here, and how much impact does it have?

Soil moisture climatological statistics from satellite retrievals and land surface models often differ, resulting partially from the physical meaning of the retrievals and the land model configurations (e.g., soil layer definitions, inputs, parameterizations, etc). These differences must be addressed via "bias correction" prior to the DA which is designed to correct random errors. Matching the mean, standard deviation, and sometimes also higher-order moments of satellite(s) and land surface model SM climatology is a commonly-used approach. See additional information from references cited here and in Huang et al. (2018). For this work, we used monthly (August) climatological statistics instead of those lumped throughout all months as in Huang et al. (2018), and a more recent version of SMAP SM data was applied. The lengthening SMAP data record and the maturing retrieval algorithm made these improvements in our methods possible.

In terms of bias correction impacts, it is mention that "*Such bias correction reduced the dynamic ranges of SM from the original SMAP retrievals*". Also note that, the used bias correction approach has shortcomings. For example, missing irrigation and other critical processes in the model can contribute to biases. If these missing processes dominantly contribute to the biases, which may not be straightforward to quantify, this bias correction approach used can remove the observational signals of these missing processes. This explanation has been added to the text.

Line 167: If soil moisture influences are not well represented in Megan, will the responses to its assimilation be meaningful or useful? The effects are only indirect through other meteorological variables.

[Figure]

The meteorological controls on BVOC emissions have been highlighted in MEGAN overview papers and have been the foci of a large number of BVOC emission studies, so the SM DA impacts on MEGAN results via changing the meteorological conditions are indeed important. Nevertheless, this SM-dependency-related limitation of MEGAN has been acknowledged and discussed around previous L395 together with the model results: "*MEGAN's limitations in representing biogenic VOC*

*emission sensitivities to SM may have had minor impacts on most of the high-isoprene-emission regions which were not affected by drought during this period*". This discussion has been extended to also cover drought-affected regions during the study period. It has been known that drought can enhance, reduce and terminate BVOC emissions, depending on the stage of the droughts and the VOC species of interest (e.g., Pegoraro et al., 2004, doi: 10.1016/j.atmosenv.2004.07.028; Bonn et al., 2019, doi: 10.5194/bg-16-4627-2019). At the early stage of droughts when plants still have sufficient reserved carbon resources, dry conditions may lead to increased BVOC emissions via enhancing leaf temperature. Persistent droughts will terminate BVOC emissions after the reserved carbon resources are consumed. Based on the PDHI maps above (source: NCDC) from July (near the beginning of the drought) to October 2016, some parts of the Atlantic states in August 2016 were in the early-middle phases of drought when reserved carbon resources were very likely still available and leaf temperature still controlled the BVOC emissions. For the drought-affected regions in August 2016, the lack of SM-dependency in BVOC emission calculations may have introduced uncertainty to the results from both the base and the SM DA cases. As SM DA only mildly affected SM and temperatures over these regions (Figure 2), we do not anticipate significant impacts of SM DA on BVOC emissions there even if their dependency on SM is realistically included in MEGAN. However, for other drought-related cases, this limitation of MEGAN may be of a larger issue, and some general suggestions on future work have been provided in the final section of this paper.

Also, please note that for this case satellite-based LAI data were used in MEGAN BVOC emission calculations. Although satellite-based LAI data may be more accurate than those calculated by dynamic vegetation models, they are less temporally-variable, and the SM DA did not adjust this MEGAN input. These also limited the responses of MEGAN BVOC emissions (and thus $O_3$ and other chemical fields) to the SM DA.

Similarly, what are the consequences of the lack of VPD treatment in the deposition scheme? This is only briefly mentioned in the text at l.400.
Impact of VPD on the stomatal resistance term of dry deposition is considered in some chemical transport modeling studies but omitted more often, as now introduced in the SI: "*A VPD limitation factor $f_{VPD}^{-1}$*

$$(f_{VPD} = min\left[1, max\left(f_{min}, \frac{(1-f_{min}) \times (VPD_{min} - VPD)}{VPD_{min} - VPD_{max}} + f_{min}\right)\right])$$

*is used in other studies to adjust the stomatal resistance term in dry deposition calculations*", referring to Chapter 3 of the Convention on Long-Range Transboundary Air Pollution, CLRTAP, 2017 (https://www.umweltbundesamt.de/en/manual-for-modelling-mapping-critical-loads-levels).

In the revision, the model-based VPD fields and their responses to the SMAP DA are now included in the SI (also shown below), together with the introductions above and additional discussions. The spatial patterns of modeled ΔVPD are shown correlated with Δtemperature fields which are anti-correlated with ΔRH.

[Figure]

It is estimated that adding a VPD limitation factor to the default Wesely scheme would decrease the modeled dry deposition velocities in the base case. For high VPD regions (e.g., >1 kPa) within our domain, depending on land cover type, this reduction may reach ~50% (referring to the $f_{VPD}$ – stomatal conductance relationships in Figure III.7 of CLRTAP, 2017). And $O_3$ concentrations may increase. Such modifications may also enhance the sensitivies of dry deposition velocities and $O_3$ concentrations to the SM DA, especially over drought-affected regions. In the manuscript, we also argue that "*the limitation would not necessarily improve the modeled deposition velocities in part due to the uncertainty in the model's LULC input and the prescribed seasonal- and LULC-dependent constants in the Wesely scheme used*". We recommend using alternative dry deposition schemes in future work, which require dynamic vegetation models. In fact, this approach, has already been tested in our Noah-MP based work which will be presented separately.

Line 169: "curves" would be clearer as "vertical profiles"
Changed to "*vertical profiles*".

Para 230: Are these observations published? If so, please provide citations.
The doi for SEAC[4]RS data has been added ("*doi:10.5067/Aircraft/SEAC4RS/Aerosol-TraceGas-Cloud*"). An earlier version of the 1-minute averaged ACT-America aircraft observations have been also archived at ORNL, with a doi:10.3334/ORNLDAAC/1593. The updated version used in this work has been posted at NASA LARC site (https://www-air.larc.nasa.gov/index.html, cited at the end of the paper).

Line 331: A table of model performance with and without DA is needed here to provide a stronger quantitative underpinning of this discussion.
Added. Please see current Table 2.

Line 343-345: There is no clear signal from the assimilation of a bias associated with irrigation in the regions indicated; why is this? Is this difference swamped by other uncertainties, or is the effect washed out by the bias correction applied before assimilation?
This description has been extended. Missing irrigation sometimes significantly affects the modeled SM which can interact with the atmosphere and introduce uncertainty to other model fields. When this dominantly contributes to the biases between the modeled SM and the satellite data, the bias correction approach applied removes that information from the satellite observations which can be an issue. In other words, this kind of bias correction approach may or may not affect the effectiveness of the SM DA over irrigated land. Enabling a reasonably-chosen irrigation scheme for the study regions which we now have tested in a different land surface model, or recalibrating the model, would help address this. While these irrigation-related issues are not resolved in this particular system, they are discussed so that the readers would interpret the DA results (absolute changes in model fields, diagnostics) over irrigated lands with caution.

Line 356-358: this explanation for model problems with evaporative fraction is vague and unconvincing!
We have extended the explanations for the model's problems with the fluxes. One major issue is related to the calibration of the C parameter in Equation (15) of Niu et al. (2011, doi: 10.1029/2010JD015139) for surface exchange coefficient ($C_H$) calculations. $C_H$ is a critical parameter controlling the total energy transported from the land surface to the atmosphere which

is directly related to the land-atmospheric coupling strength. The default value of C=0.1 is used in the Noah land surface model to derive roughness lengths in the $C_H$ calculations, which may be highly unrealistic. Some previous studies have concluded that C may be underestimated by a factor of 5 in some environments/periods, resulting in significant biases in modeled energy fluxes which cannot be resolved solely by adjusting the modeled soil moisture and vegetation fields (LeMone et al., 2008, doi: 10.1175/2008MWR2354.1). Ideally, C should be calibrated for various land cover types or canopy heights based on observations. We also recommend using alternative $C_H$ parameterizations that are available in other land surface models. As demonstrated in previous studies, more accurate model calculations of $C_H$ would also benefit the partitioning of water fluxes (evapotranspiration vs runoff) in the land system, as well as predicting the weather conditions.

A second flux-related weakness of the modeling/DA system used is that vegetation and albedo in Noah were not updated by the SM DA, which is unrealistic. Additionally, we pointed out that the modeled soil states and fluxes are sensitive to soil parameters (dependent on soil type and a look-up table) in Noah which may not be up-to-date. We anticipate that improving the $C_H$ scheme and assimilating SM alone or together with other land observations into dynamic vegetation models (with up-to-date soil/vegetation parameters) will help address such flux problems and also further improve the weather states. Our initial results based on different $C_H$ treatments and the dynamic vegetation option in the Noah-MP model, which will be shown in a separate study, confirmed these explanations.

Line 367: The impacts of the data assimilation on temperature and humidity are very small. Are these changes significant?
A set of figures (also shown below) has been added to the SI based on student's t-tests for modeled 2 m air temperature, RH and 10 m wind speed from different cases. The readers may use these results to interpret the absolute model responses to the DA, keeping in mind that the assumptions of student's t-test are not always met. For air temperature and humidity along flight paths, we show model performance at various flight altitudes in Figures 6-7. In the text of this section, we mention not only the overall statistics but also the maximum changes in air temperature and humidity along flight paths which are not small. According to this referee's other comments, we added sensitivity studies for selected flight days to better explain where intense changes occurred during individual events and why.

[Figure]

Fig 6 shows the observations, the model simulation and the impacts of assimilation. However, it does not show whether the base simulation matches the observations or whether the assimilation improves the model bias, and these are the two factors that the reader is most interested in! Some

of this information is provided in Figure 7 on a temporal not vertical basis, but please reconsider which panels to show in Fig 6.

The impacts of SM DA on the model fields and model performance are different and both of them are informative. Figure 6 has been reorganized with the addition of model performance changes as a function of flight altitude. RMSEs and their changes are also summarized by flight altitude in Figure 7.

Line 405: It would be worth pointing out that these RMSE changes are positive and that model performance is less good with assimilation.

The sentence has been reworded, and "*increased*" is used before these positive numbers instead of "*changes*". This more clearly indicates that the overall model performance was slightly degraded.

Line 426: The points made in this paragraph highlight compensating model errors for ozone, but the lack of any stratospheric influence in the WRF-Chem runs remains an issue to be addressed.

Agreed.

Line 451: lightning is mentioned in the abstract, conclusions and a number of places through the paper, but the effects are not quantified anywhere. Does the soil moisture assimilation have any significant effect on lightning NO emissions? If so, please quantify it.

The passive lightning $NO_x$ tracer was implemented in all WRF-Chem simulations. We now include the visualizations of the lightning $NO_x$ tracer results in the SI and discussed them in the text. The temporally-averaged results are also shown below. Also, this conclusion is drawn after comparing the modeled intra-cloud and cloud-to-ground flash counts from various simulations.

[Figure]

Line 495: Evaluation against SEAC4RS observations is not thorough here. Assimilation "led to better model agreements" but no numbers are provided in support of this. Some indication of the biases or RMSE values are needed in the text or a table, or alternatively a scatter plot of simulations against observations should be added to Figure S5. While this attempt to put the results of the study in context is valuable, the comparison is not convincing, and the explanations are highly speculative.

In Figure S10 (previous Figure S5) and its caption, we now report model evaluation results (based on RMSE and correlation coefficient metrics) with various observational datasets collected during SEAC[4]RS. We use different colors in the "DA-no DA" plots to indicate whether the SM DA improved or degraded the model performance. In the text, the specific locations where the SM DA had notable positive impacts on WRF-Chem simulations are highlighted.

Line 510: Improvements in T2/RH/WS in 50% of locations is not a convincing demonstration of the value of assimilation. The improvements in MDA8 against AQS and CASTNET (42%, 51%) are of very similar (negligible?) magnitude, but these details are omitted from the concluding discussion.

The descriptions of model performance changes have been modified in the conclusion discussion. RMSEs are referred to.

Fig S1: The panels in this figure are too small, please make them larger so that they are legible (as in Fig 1).
The orientation of this set of figures has been changed to significantly improve its readability.

Typos and Minor Issues
The language needs substantial polishing, e.g., line 103 "of the used modeling system" better as "of the modeling system used". (and Line 341)
Done, and also applied to similar language throughout the manuscript.

Line 111: acronym SRTM30 is not defined.
SRTM30 is now spelled out as "*Shuttle Radar Topography Mission Global Coverage-30*".

Line 139: is -> are
Done.

Line 340: better phrased more clearly without use of "unmodeled"
The word "unmodeled" appeared more than once in the previous version of the manuscript, and it has been replaced by "*missing processes*" or "*unaccounted for*" depending on the context.

**Response to RC2**

This study addresses the impact of a more accurate treatment of soil moisture content on WRF-Chem simulations of some aspects of weather and atmospheric composition. The soil moisture of the NOAH land surface model is adjusted using a data assimilation technique to retrieved soil moisture content from the NASA SMAP radiometric measurements. The more accurate soil moisture then modifies moisture, heat and trace gas emissions from their 'base' values, and two WRF-Chem studies are compared for the period of August 2016, one being the base and the other including SM DA. Additional simulations are performed for 2013. Comparisons are between ground and aircraft-based observations and modelled quantities.

This is a competent modelling study, and the authors have attempted to apply best practice in bringing SM DA and the application of WRF Chem to the study of the continental US. As such it complements, but doesn't much extend, an earlier study by the lead author in 2018. It therefore somewhat lacks novelty.

The SM DA is shown to improve model performance as compared to aircraft observations of air temperature and specific humidity, although there is no reported improvement against ground-based observations of temperature, humidity or wind speed. For reactive gas phase composition, very small changes in O3 are calculated, with little effect of SM DA on modelled ozone aloft. Some degradation in model skill results, which the authors phrase as being less 'desirable'. I think this means they expect SM DA to improve model skill, but as it stands there are no reasons in the manuscript given.

The study concludes with the effect of including an updated emissions database on modelled ozone.

I would identify this as an interesting region/time period for study being a geographical region with a heterogeneous LULC environment results where there are multiple sampling of edge cases (regions of drought, regions close to field capacity) in the the vegetation modelling framework.

My main issue with the MS is that it's something of a pot-boiler, and the problem under consideration is not clearly stated. The study is undermined by the majority of the discussion being rather qualitative, despite much quantitative information being in the paper's figures, and the discussion is often focussed on what was not included, rather highlighting the impact of SM DA on model performance. Some important questions are raised, but no clear direction of travel for this work emerges and the no clear conclusions are drawn as to how and to what extent SM modifies the picture until we reach the concluding remarks. This diminishes its impact and I suggest that the focus of any revised submission should be on the process-level impacts of SM DA on e.g. emissions or deposition processes which result from the improved treatment of SM.

Thank you for the overall positive feedback as well as the suggested changes. Compared to Huang et al. (2018) that this referee mentioned about, this study focuses on a different region (southeastern US), different time periods (summer convective season, specifically, during two field campaigns in August 2016 and August 2013, respectively), as well as different chemical species ($O_3$ and its precursors). As also mentioned in our response to RC1, the SM DA and bias correction approach were improved to some extent, benefiting from the lengthening SMAP data record and the maturing SMAP retrieval algorithm. We also address some of the limitations brought up in Huang et al. (2018), such as the uncertainty in bottom-up anthropogenic emission inventories, and lack of evaluation of modeled fluxes. This study also reveals multiple major shortcomings of the widely-used Noah land surface model in studying the SM interactions with weather and atmospheric chemistry, and discusses the improvements that we may expect from Noah-MP based experiments (to be shown separately).

The model performance for surface air temperature, humidity, wind speed and $O_3$ is summarized in Table 2 of the revision. The model evaluation with aircraft observations has been extended, and sensitivity analysis has been added to compare and contrast the significantly different conditions during two ACT-America flight days at the surface and aloft. We strongly recommend looking at these results event by event, in addition to referring to the overall statistics. In general, "desirable" refers to improved model performance for the variables of interest due to any possible reasons. In both the main text and the SI, we extended the analysis and discussions related to emissions and deposition calculations as well as their connections with the model performance changes.

I say this because, at present, the authors bring up aspects of the modelling framework which are unsatisfactory or where the study itself could have been improved and some readers might be left wondering exactly what remains of the SM impact that has been included at the process level. Slightly frustratingly, there are long parts of the discussion on things that can't be addressed (340-345 irrigation. 346-347 rainfall product QC, 358-360 other models that might be used). L210 raises the question of how SM affects convection, but no discussion of the impact is given on e.g. vertical transport.

The Noah land surface model used in this study has long been, and is still, widely used in land, weather and air quality modeling communities. Therefore, we believe that case studies using Noah

are informative to various audiences. Many air quality modeling applications based on Noah and other models do not (suitably) include irrigation. Missing irrigation sometimes significantly affects the modeled SM which can interact with the atmosphere and introduce uncertainty to other model fields. It can also affect SM bias correction. It is very important to bring up the limitations related to irrigation as well as the uncertainty in evaluation datasets (e.g., precipitation data) so that the readers can take the results with caution. To address some of these limitations, the utilizations of other land surface models or/and WRF physics would be needed.

Downward and upward movements of air pollutants are discussed further in the revision. The analysis of WRF-Chem modeled vertical wind speed (W) is now included in the SI and shown below as well. Also, like in many studies, CO is used primarily as a tracer of transport, and their distributions are investigated at event- and 13-day timescales.

[Figure]

As a second example, on L167 the critically important aspect of the response of MEGAN to SM is raised, but, after saying that SM effects are not well understood, the discussion moves on, although figure 3 shows the modifications. The authors need to extend this section extensively to quantify the effect of SM on emissions in MEGAN, and to show how MEGAN responds (especially as these NO and isoprene emissions changes are important to the discussion of the ozone response to SM DA).
The MEGAN-related paragraphs have been modified significantly accounting for both referees' comments. Please also see detailed information in page 5-6 of this document.

As a third, L187 raises SM-dependent vegetation properties which might be important to ozone deposition, but it is again not clear what the impact of these effects might be on e.g. ozone. These issues are raised again L379 and again in L396-403 with similar lack of clarity as to their impact. If these important effects can't be at present included, it seems to me that having raised these points repeatedly, the authors should at least estimate what the size of these effects might be on deposition velocity and hence ozone flux to the surface? The manuscript would be improved drastically if these important processes were discussed quantitatively - above, it would be good to know how big are the emissions changes. Here, how large would deposition velocities need to change to produce an effect on ozone?
The analysis and discussions regarding dry deposition have been extended-please also see detailed information in page 6-7 of this document. We stated in the revision that "*If the SM and VPD limitation factors (details in the captions of Figures S1 and S7) were included in the calculations, the modeled deposition velocities in both the base and the "assim" cases would become smaller, and the SMAP DA may result in more intense relative changes in the modeled deposition velocities, especially over drought-affected regions. Including such limitation factors, however, would not necessarily improve the modeled deposition velocities in part due to the uncertainty in the model's LULC input and the prescribed seasonal- and LULC-dependent constants in the Wesely scheme used*".

Please note that, by applying the SM and VPD limitation factors, the relative decreases in dry deposition velocity may intensify in places, associated with $O_3$ enhancements. Previous studies that evaluated the Wesely scheme with flux observations reported net underpredictions in dry deposition velocity for most land cover types except cropland. Several references cited in this paper show that via updating the Wesely scheme with physiological scheme for stomatal resistance, dry deposition velocity increased by ~0.2 cm s$^{-1}$ and surface $O_3$ decreased by ~7 ppbv over the southeastern US during the summertime of other years. These updates in the dry deposition schemes effectively reduced the positive biases in modeled $O_3$. Similar modifications have been applied in our Noah-MP based modeling experiments. During the ACT-America 2016 period, we see that the increases of 0.1-0.2 cm s$^{-1}$ in dry deposition velocity over some non-cropland regions led to up to ~3 ppbv decreases in mean surface $O_3$; and due to this update, dry deposition velocity over cropland decreased by 0.02-0.05 cm s$^{-1}$, and the resulting $O_3$ enhancements are mostly <0.5 ppbv. These results will be presented and discussed separately. Partially based on the references and additional experiments, in this paper, the regions experienced strong changes in dry deposition velocity are highlighted, linked with the $O_3$ changes: "*..These responses are within ±0.02 cms$^{-1}$ in >70% of the model grids but are outside of ±0.05 cms$^{-1}$ in Ohio and Missouri where they were highly responsible for the surface $O_3$ changes*".

The maximum MEGAN emission responses to the SMAP DA (relative to the base case, in %) are now specified in Section 3.3, occurring over the regions where daytime surface $O_3$ reacted most strongly. And yes, $O_3$ enhancements over some of these regions are also due to the reduced dry deposition velocity as mentioned above. We have clarified that in the Noah model vegetation and albedo were not updated with SM, which also affected the responses of surface fluxes, weather conditions, as well as biogenic emissions. We suggest that these limitations may be addressed in the future by applying a different land model with dynamic vegetation and multivariate land DA which would also benefit the dry deposition calculations.

As a fourth example, the study makes a point about how the signal from the use of different emissions datasets in terms of ozone response is large. This is a worthy point, but does not contribute to the question at hand, and the use of a second emissions dataset doesn't really improve the understanding of the problem. Similarly, the role of strat-trop transport is undoubtedly important, but again moves the discussion away from the SM DA. The reluctance to exclude anything, and to state which factors are dominant, makes the focus of the study very difficult to discern, and really detracts from the potential impact which is to understand how the WRF-Chem modelling framework is improved by SM DA in this mixed LULC environment.

Both referees have recognized that a main aspect of this study is quantifying SM impact on model performance. As shown in the paper, anthropogenic emissions and strat-trop transport exert strong controls on WRF-Chem $O_3$ error budgets throughout the troposphere, and thus they significantly affect the assessment of SM DA impacts on the modeled $O_3$ performance. These have already been emphasized in the abstract and multiple sections of the manuscript. Please note that, currently, NEI 2014 is used in many modeling studies for periods after 2016, scaled by no or constant factors. By demonstrating the benefit of using NEI 2016 beta, we stated that "*using up-to-date anthropogenic emissions in WRF-Chem would be necessary for accurately assessing SM DA impacts on the model performance of $O_3$ and other air pollutants*". Although NEI 2016 beta is developed with the base year of 2016, it is still important to "*..continue to improve NEI 2016 beta and any newer versions of emission estimates...*"

Additionally, we have tightened the connections of anthropogenic emissions and other emissions, as well as convection/cold fronts and strat-trop transport. For example, modeled $O_3$ responses to biogenic emissions, which are sensitive to weather/SM, also depend on the model's anthropogenic emissions inputs (see several references cited in Section 3.1). Stratospheric intrusions are often associated with cold fronts or/and convection, as demonstrated in numerous previous studies some of which are cited (e.g., Ott et al., 2016; Pan et al., 2014, doi: 10.1002/2014GL061921, based on evidence from other field studies/models). During our study periods, stratospheric $O_3$ influences that were observed on the B-200 aircraft and/or modeled by global modeling systems (e.g., AM4) were also possibly linked to convection initiation and development, lightning and its emissions, vertical/horizontal transport which are sensitive to SM. With this being said, it remains challenging to accurately simulate these processes in both coarse-resolution global models and regional models like the WRF-Chem system used here.

The ozone response appears to be driven by temperature via the coupling of MEGAN to WRF meteorology. Here the manuscript is somewhat successful but this section also gives the clearest indication of how it could be improved. The authors could expand on the description of the results to drill down into the processes at work and how they interact. For instance, Figure 9 shows a very small change in ozone, which receives little comment, the authors preferring instead to concentrate on the maximum value and the correlation. The revised MS could look at regions of positive or negative ozone change, and say whether the small change in O3 is to be expected, or not, and give reasons for this, for instance by unpicking the contribution from emissions, deposition and temperature changes in Figure 3, and to present the results in more detail than is done in L389-392. This approach should be followed for the other aspects of the impact of SM DA on O3 and other atmospheric constituents.

Please see responses to this referee's previous comments, particularly the second and third general comments.

In conclusion, I feel that the impact of the study would be improved if the focus could be narrowed, the depth of discussion improved and the connection of SM to the other inputs to WRF-Chem better quantified.

As specified in our previous responses, the paper has been reformulated. Additional analysis and discussions are included; the connection of SM with other factors affecting model performance has been tightened; and some of the materials have been removed or moved to the SI.

Specific comments:

L43: missing symbol between 70 ppbv

We assume that this comment suggests that "ppbv" should be spelled out here. The text has been changed to: "..*parts per billion per volume (ppbv..*".

L55: mid-latitude cyclones are and L56: 'They are'

Done.

L138-139: 'the major chemical species in the FT are'

Done.

L148: what do the authors mean by 'runs'?

Changed to "*simulations*".

L180: rephrase 'its major component surface resistance'
Changed to: "*Over land, surface resistance, the major component of dry deposition velocity,…*"

L307: shown to be consistent
Done.

L330: unusual use of dominantly
Changed to "*prevalently*".

L533: sentence describing the impact is not clear
This sentence now reads as: "*The impact of SMAP DA on upper tropospheric $O_3$ was partially via altering the transport of $O_3$ and its precursors from other places as well as in-situ chemical production of $O_3$ from lightning NO and other emissions (including $O_3$ precursors transported from elsewhere).*"

**References and Acronyms**

The dois for added references are provided in this document. The full citations for all references are available in the revised manuscript and its SI. Acronyms in this document are also defined in the manuscript.

---

## Referee Report (RR1)

I have two main issues:

First, the revised MS does not address my comments substantively. The revised MS shows only a few significant changes. I do not consider that the structural problems in the MS raised in the review have been addressed - the focus is on model response but there is not enough time spent quantifying at the process level the impact of DA. Second, there's been no restructuring to tighten up the focus to the subject mentioned in the title, and so this remains a paper where the apparent subject of the paper is lost in the details of comparison between model and observations and the simple sensitivity studies that would give a much better process-level understanding of the impacts of DA of SM on model performance are not there.

In lots of places the discussion in the manuscript remains rather qualitative. The revised MS spends too much time discussing factors not included in the study (e.g. L532-538, discussion of possible modifications/missing processes in O3 deposition), and too little time is spent on discussion of the model itself (e.g. L529-L530 which is all that's said on the results regarding O3 deposition). I'd suggest removing as much as possible of this speculation about model inadequacies that does not add to the interpretation of model response, and refocusing the MS by adding a separate detailed section on what processes are susceptible to modification by SM DA and on quantifying the change in these at the process level (that is, for instance, at the emissions level rather than on the impact on the very small change in ozone levels).

A MS that had a section on 'model responses to SM DA' moving figures S3 and S4 into the main text and then a section on 'comparision with observations' would allow the reader to assess better how the impact of SM DA propagates through to model skill in simulating ozone. Then add significant extra text to 'model responses to SM DA' section in the MS discussing these S3 and S4 figures and similar process level responses of model inputs or other parameterisations before considering the impact on O3.

Consider L220 of the revision where extra text has been added. The revised discussion is not based around how the biogenic emissions varied with the assimilated SM conditions in this study, or even quantifies the difference. In L515 finally the authors state that the MEGAN emissions have no dependency on SM, but again 'do not anticipate' that emissions were changed. This is not a sufficient response to the review. Please insert a quantitative discussion as to how SM DA affected, and why, the biogenic VOC and NO emissions in your study.

Consider, as a second, but not final example, dry deposition, L240/L535 again does not make clear how the deposition velocity is affected by SM DA - is it solely via modification to surface temperature? Again, it would help to use the Weseley scheme to estimate the response of the deposition velocity to the temperature. This would help to understand the S4 figure panel on deposition. At present, deposition has only been addressed with extra text stating qualitatively how things might be (pg L221, L516, L535), but the calculations/diganostics are not there. Again, what drives the change? Again, please add a quantitative discussion as to how SM DA affected deposition.

Similar comments pertain to my other major points where I asked for quantitative discussion. In a study such as this where you are trying to unpick how SM DA affects O3, these process-level attributions are essential to the success of your study.

---

## Author Response (AR2)

**Author response to reviews**

The authors appreciate ACP Editorial Office, Dr. Müller, and both referees' continued efforts. These elaborated comments helped us further improve this manuscript significantly. Following Dr. Müller's guidance, we paid particular attention to the feedback in Report #2 when revising the manuscript. Please see below our point-by-point response (in blue) to all comments by the ACP Editorial Office and both referees (in black). Quoted text from the revised manuscript is *in italic*. A "tracked-changed" version of the manuscript is submitted together with this document.

**Response to ACP Editorial Office comments**

I noticed that your table 3 contains coloured cells. Please note that this will not be possible in the final revised version of the paper due to HTML conversion of the paper. When revising the final version, you can use footnotes or italic/bold font.

In Table 3, degradations and improvements are now highlighted in italic and in bold, respectively. Regarding the paper format, the titles and footnotes of tables have been carefully edited, and the unit of temperature has been changed from "oK" to "K" in the text, Table 2, as well as several figures in the manuscript and the SI.

**Response to Report #1 by Referee #2**

The authors have addressed the reviewer comments reasonably well, revising the text, adding a useful additional simulation and refining their explanations. The resulting manuscript is an improvement over the original, and a better account of their results. The perspective remains rather narrow, focusing more on documenting results than on furthering understanding. This seems like a missed opportunity, but the paper nevertheless presents some useful results that are worthy of publication.

Thank you for recognizing the improvements in this manuscript. As this referee pointed out in the earlier review report, the narrow perspective is partially due to the choice of the Noah land surface model. This will be better addressed by presenting results based on a different land surface model in a follow-up study. To improve our understanding of how SM DA affected ozone at process level based on the widely-used Noah model, especially at/near the surface, additional analysis has been conducted, the results sections in the manuscript and the SI have been reorganized, and the abstract has been rewritten. Please refer to Figure R1 in this document for a clear illustration of SM DA impacts on modeled ozone and its performance at process level which are discussed in the revised paper. Please see our responses to your next two comments for the changes made to the abstract and the reorganizations of results.

The abstract describes what was done, but still doesn't summarize the results of the study in a meaningful way beyond stating that SMDA impacts the model results. Not only is there no quantification, the statements are barely even qualitative. The only conclusive statement is the new comment that "SMDA improved the model treatment of convective transport" (although the "and/or" that follows suggests that even that is uncertain). The abstract should state clearly what the study found.

It would be helpful to have an overarching conclusion that the reader can take away (that SMDA doesn't make much difference given other biases?) The conclusions quickly get into technical detail and explanation, and the reader loses the big picture.

Please note that unlike aircraft CO measurements which were used to assess the model treatment of transport during both the ACT-America and SEAC4RS campaigns, aircraft NOx measurements which helped evaluate WRF-Chem lightning NOx at higher altitudes were available only during the SEAC4RS campaign. This is a reason for using "and/or" in that sentence.

The abstract has been rewritten, including quantitatively descriptions of key ozone-related findings and other highlights. Following this referee's suggestion, we use the following sentence to deliver an overarching conclusion of this study, and to emphasize the strong needs of continuing such studies with improved model parameterizations (e.g., deposition) which will be covered in a follow-up study: "In the cases that the DA improved the modeled SM, weather fields and some O3related processes, its influences on the model's O3 performance at various altitudes are not always as desirable, due to the uncertainty in the model's key chemical inputs (e.g., anthropogenic emissions), as well as the shortcomings in model parameterizations (e.g., chemical mechanism, natural emission, photolysis and deposition schemes) and the model representation of stratosphere-troposphere exchanges".

Many of the changes have involved adding additional material to the supplement. While this makes the record of the results more complete, it doesn't help the reader focus on the key aspects that matter.

Some of the previous SI contents (i.e., previous Figures S3-S4) have been moved into the main manuscript, and additional contents (i.e., current Section S2, Figures S4 and S6, two new panels in current Figure S5, and a new column in Table 2) have been added to support the extended discussions.

Line 99: The term "semicoupled" is still not adequately defined. Remove "here is similar to weakly-coupled as opposed to fully or strongly coupled, which" as this does not help the reader at all. The remainder of the sentence at least states what is coupled.

The cited text has been removed. This paragraph introduces the paper organization, whereas the following "methods" section 2.1 includes most of the technical details, e.g., regarding the coupling, please see the paragraph starting with "*All WRF-Chem cases, except case "minus001*…".

**Response to Report #2 by Referee #1**

I have two main issues: First, the revised MS does not address my comments substantively. The revised MS shows only a few significant changes. I do not consider that the structural problems in the MS raised in the review have been addressed - the focus is on model response but there is not enough time spent quantifying at the process level the impact of DA. Second, there's been no restructuring to tighten up the focus to the subject mentioned in the title, and so this remains a paper where the apparent subject of the paper is lost in the details of comparison between model and observations and the simple sensitivity studies that would give a much better process-level understanding of the impacts of DA of SM on model performance are not there.

We appreciate the elaborated comments and suggested changes by this referee. To improve our understanding of SM DA impacts on the modeled ozone at process level, especially at/near the surface, additional analysis has been conducted, results sections in the manuscript and the SI have been reorganized (please refer to our responses to your following comments for details), and the

abstract has been rewritten. Figure R1 below clearly illustrates the SM DA impacts on modeled ozone and its performance at process level which are discussed in the revised paper.

---

## Author Response (AR3)

**Author response to reviews**

The authors appreciate ACP Editorial Office, Dr. Müller, and both Referees' continued efforts. Following Dr. Müller's guidance, we further improved the paper along the lines suggested by the Referees. The changes include adding "take-home" messages in the conclusions as well as revising related sentences in the abstract. Please see below our response (in blue) to Referee #2's remaining comments (in black), which are also in line with Referee #1's general suggestion regarding making "the most of the opportunity afforded to improve our understanding of soil moisture on air quality". Quoted text from the revised manuscript is *in italic*. A "tracked-changed" version of the manuscript is submitted together with this document.

**Response to Report #1 by Referee #2**

The authors have made changes to the manuscript to address the comments of both reviewers, although these are not particularly convincing. The paper still suffers from the key problem that I identified in my original review: that it is descriptive rather than analytic. It is a very competent report of the research performed, but it remains short on the type of new insight needed for a useful scientific paper. There is a lot of explanation and justification of model performance, but it is not clear what we learn from this. The topic is certainly of interest to the readership of ACP, and worthy of publication, but the reader is made to work hard to extract the key messages of the study. The research covered by this paper aims to bring in new insights into the added value of assimilating satellite soil moisture data to modeling atmospheric conditions in a regional-scale coupled modeling system comprising the widely-used Noah land surface model. The "atmospheric conditions" here not only refer to weather fields (which have been topics of many previous studies) but also air quality related states and processes, as noted in Section 1 of the paper. Explanations and justification of the model behaviors are necessary for understanding these Noah-related results. They also help lead to the expectation that the value of soil moisture data assimilation would be larger in modeling systems where soil moisture-sensitive processes are more realistically represented, and other sources of model uncertainty are better addressed. Including key messages in the conclusions, as this referee suggested below, also help address this comment.

I requested a clear, overarching conclusion for the reader to take away. The response is a lengthy sentence in the abstract that lists uncertainties but provides little useful information to highlight the take-home messages. There are minor adjustments to detail in the conclusions, but these provide no overview of the bigger picture that might highlight the wider value of the study. This still needs to be resolved. The final sentence of the abstract advertises future work, but this appears to underscore the rather inconclusive nature of this study, which needs to stand on its own.
That added sentence has been broken down into three sentences and reworded. Now this part reads as: "*In the cases that the DA improved the modeled SM, weather fields and some $O_3$-related processes, its influences on the model's $O_3$ performance at various altitudes are not always as desirable. This is in part due to the uncertainty in the model's key chemical inputs, such as anthropogenic emissions, and the model representation of stratosphere-troposphere exchanges. This can also be attributable to shortcomings in model parameterizations (e.g., chemical mechanism, natural emission, photolysis and deposition schemes), including those related to representing water availability impacts.*"

The outcome from this work, based on the Noah land surface model, can stand on its own, although it's also noted that the value of soil moisture data assimilation would be larger in modeling systems where soil moisture-sensitive processes are more realistically represented and other sources of uncertainty of the model are better addressed. The concluding paragraph has been modified to more clearly highlight the key messages:

"*...It was demonstrated that, via changing the model's weather fields that drove its chemistry calculations online, the SM DA influenced various $O_3$-related processes, $O_3$ concentrations and exceedances modeled by WRF-Chem. In some locations/times, these influences were large and resulted in improved model performance. To further improve the modeled chemical fields via applying the SM DA at various scales, it is not only important to improve the model representations of anthropogenic emissions and trans-boundary transport, but also to address shortcomings in model parameterizations, e.g., to realistically reflect the impacts of water availability on biogenic emissions and dry deposition, and for longer simulations, to include $O_3$ damage to vegetation…*"

The descriptions in the text would still benefit from some polishing to make them clearer and thus more accessible to the reader. For example, on lines 287-8, the synoptic conditions and drought conditions "can be closely linked to" regional O3 variability. I assume the intended meaning here is "partly explain"? There are a number of places like this where some relationship is identified but where the direction or magnitude of the relationship isn't clear.
We have changed this text to "*partially explain the regional $O_3$ variability*".